# Personalized risk stratification in colorectal cancer via PIANOS system

Du Cai [1,2,3,9], Haoning Qi [1,2,3,9], Qiuxia Yang [4,9], Huayu Li [5,9], Chenghang Li [6], Chuling Hu [1,2,3], Baowen Gai [1,2,3], Xu Zhang [7], Yize Mao [8] ✉, Feng Gao [1,2,3] ✉ & Xiaojian Wu [1,2,3] ✉

Current prognostic biomarkers for colorectal cancer (CRC) lack stability and generalizability across different cohorts and platforms, challenging precise patient stratification. Here, we introduce a Platform Independent and Normalization Free Single-sample Classifier (PIANOS), designed to refine treatment decisions by accurately categorizing patients with CRC into distinct risk groups. Developed using gene expression data from 562 patients and employing a rank-based k-Top Scoring Pairs (k-TSP) algorithm alongside resampling, PIANOS was rigorously validated in 15 cohorts comprising 3666 patients with CRC. It effectively differentiates high-risk from low-risk patients, outperforms 105 existing models, and demonstrates robust performance across technologies like microarrays and RNA sequencing. PIANOS-based stratification is validated as an independent predictor of disease-free survival. Moreover, PIANOS discriminates treatment responses across risk categories, with high-risk patients showing increased sensitivity to bevacizumab and low-risk patients exhibiting enhanced responsiveness to chemotherapy and immunotherapy. This study reports significant advancements in supporting clinical decision-making for CRC and provides a reliable framework for optimizing patient treatment strategies.

Colorectal cancer (CRC) is the third-most common malignancy and the second-leading cause of cancer-related death worldwide[1]. Despite considerable progress in the therapeutic approaches for CRC, patient outcomes remain unsatisfactory, with a general five-year survival rate of ~65%[2]. A robust, accurate, and clinically actionable risk stratification system provides a foundation for further drug discovery to improve CRC patient outcomes[3]. Since not all CRC patients at the same stage respond consistently to the same treatment, a considerable number of patients may experience overtreatment, undertreatment, or receive no effective treatment at all[4,5]. Therefore, accurate and robust risk stratification would be crucial to improving prognostic outcomes for patients with CRC.

[1]Department of General Surgery (Colorectal Surgery), The Sixth Affiliated Hospital, Sun Yat-sen University, Guangzhou, P. R. China. [2]Guangdong Provincial Key Laboratory of Colorectal and Pelvic Floor Diseases, The Sixth Affiliated Hospital, Sun Yat-sen University, Guangzhou, China. [3]Biomedical Innovation Center, The Sixth Affiliated Hospital, Sun Yat-sen University, Guangzhou, P. R. China. [4]Department of Radiology, State Key Laboratory of Oncology in South China, Guangdong Provincial Clinical Research Center for Cancer, Sun Yat-sen University Cancer Center, Guangzhou 510060, P. R. China. [5]State Key Laboratory of Oncology in South China, Guangdong Provincial Clinical Research Center for Cancer, Sun Yat-sen University Cancer Center, Guangzhou 510060, P. R. China. [6]Artificial Intelligence Thrust, The Hong Kong University of Science and Technology, Guangzhou, China. [7]Center for Reproductive Medicine, Chongqing Reproductive Genetics Institute, Chongqing Health Center for Women and Children, Women and Children's Hospital of Chongqing Medical University, Chongqing, P. R. China. [8]Department of Pancreatobiliary Surgery, State Key Laboratory of Oncology in South China, Guangdong Provincial Clinical Research Center for Cancer, Sun Yat-sen University Cancer Center, Guangzhou, P. R. China. [9]These authors contributed equally: Du Cai, Haoning Qi, Qiuxia Yang, Huayu Li. ✉e-mail: maoyz@sysucc.org.cn; gaof57@mail.sysu.edu.cn; wuxjian@mail.sysu.edu.cn

Recent notable advancements in CRC diagnosis and treatment include identifying biomarkers like deficient mismatch repair (dMMR) and microsatellite instability-high (MSI-H), which have proved instrumental in guiding immunotherapeutic strategies[6]. However, these biomarkers are present in only about 15% of CRC cases, limiting their broader applications[7]. Further, most clinical studies still rely predominantly on the TNM staging system, which accurately gauges tumor burden but fails to capture tumor heterogeneity[8]. Consequently, the effectiveness of numerous CRC clinical trials is hampered.

Technological advancements have introduced various risk stratification methods for CRC, such as whole-exome sequencing, RNA sequencing, immunohistochemistry, radiomics, and pathology[9–13]. For instance, the pathology-based SIA[14], the iCMS developed through the integration of single-cell and transcriptomic analyses[15], and the CRPS—which converts transcriptomic data into pathway enrichment scores and uses deep learning for modeling—[16]have all been explored. However, these biomarkers frequently face accuracy and stability issues due to variability in detection methodologies, sequencing platforms, and patient demographics[17]. Such issues limit the applicability of gene expression-based biomarkers to their initial training sets and the selection of public datasets. Further, normalization requirements for dataset processing often introduce test set bias, which refers to additional information introduced during the data standardization process, hampering accurate prediction for new individual samples[18,19]. Even widely adopted solutions, such as a combat algorithm, fail to completely negate batch effects[20].

In this work, we present PIANOS, a robust, platform-agnostic classifier for stratifying CRC patient risk. We demonstrate its prognostic performance across multiple cohorts and its ability to identify distinct treatment sensitivities. Specifically, PIANOS suggests that low-risk patients may benefit more from chemotherapy and immunotherapy, whereas high-risk patients may exhibit activated angiogenic features. This study underscores the potential of PIANOS to enhance personalized clinical decision-making in CRC.

## Results

### Construction and evaluation of PIANOS in CRC cohorts

Figure 1 provides an overview of the research process. We analyzed data from 24 cohorts across ten countries comprising 5439 patients (Fig. 2A). We used the single-sample gene set enrichment analysis (ssGSEA) algorithm to compute enrichment scores for 22,596 pathways from the MSigDB database (version 7.0) for each patient. PIANOS was developed using k-TSP and resampling algorithms, training with 364 patients from CIT (Fig. 2B). PIANOS stratified patients into high- or low-risk categories based on gene expression profiles, designating those scoring >17 as high-risk. Baseline information for all major cohorts is presented in Supplementary Data 1–3. We examined the distribution of risk scores across different T, N, and M stages in COCC and other cohorts, finding a significant increase with advancing cancer stage (Fig. 2C and Supplementary Fig. 1A–C). Multivariate analysis for DFS revealed that PIANOS stratification operates independently of TNM staging and MSI status (Fig. 2D and Supplementary Data 4–9). Subsequently, DFS disparities between high- and low-risk groups across various cohorts were investigated. DFS in high-risk patients was considerably shorter compared to those in the low-risk group across all examined cohorts (Fig. 2E–I). Besides, PIANOS effectively stratified patients with CRC across different TNM stages (Supplementary Fig. 1D–F). Supplementary Fig. 2A to C illustrates the distribution of KRAS mutations, BRAF mutations, and tumor locations across the high- and low-risk groups in the COCC cohort. PIANOS effectively stratified patients according to prognosis across different Clinicopathological characteristics (Supplementary Fig. 2D–L). Figure 2J, K presents typical computed tomography (CT) images from different risk groups, highlighting cases in which a low-risk patient with advanced tumors exhibited longer DFS than a high-risk patient with early-stage tumors,

who experienced early recurrence. This demonstrates PIANOS' ability to discern tumor biological behaviors beyond traditional TNM staging, offering a refined approach to CRC risk stratification.

### Performance and robustness of PIANOS

Figure 3A shows the distribution of patients across high- and low-risk groups within all analyzed cohorts, as detailed in Supplementary Data 10. We evaluated the prognostic capability of PIANOS by computing the concordance index (C-index)[21] and D-index[22] for each cohort. The analyses confirmed its ability to stratify patient outcomes overall (overall C-index = 0.74, 95% CI = 0.71–0.76; overall D-index = 2.95, 95% CI = 2.56–3.34; Fig. 3B, C and Supplementary Data 11, 12), considering data from both sequencing (C-index = 0.71, 95% CI = 0.61–0.81; D-index = 2.49, 95% CI = 1.58–3.41) and array (C-index = 0.75, 95% CI = 0.71–0.78; D-index = 3.10, 95% CI = 2.62–3.58) platforms. To further evaluate PIANOS's efficacy in predicting recurrence, we calculated the area under receiver operating characteristic curve (AUROC) for each cohort independently. The results consistently demonstrated significant predictive power for recurrence across all cohorts (Fig. 3D–H and Supplementary Fig. 3A–E). To compare the predictive performance of PIANOS with that of existing colorectal cancer prognostic systems, including SIA, iCMS, CMS and CRPS, we evaluated the ROC curves for 3-year and 5-year recurrence across different cohorts (Supplementary Fig. 3F–O), revealing that PIANOS exhibited the highest predictive capability in all cohorts. To further contextualize our findings, we performed survival analysis on the different subtypes defined by the CMS, SIA, iCMS, and CRPS classification systems within the COCC cohort. Importantly, the prognostic stratification observed within our COCC cohort for these classification systems was consistent with their previously reported prognostic performance (Supplementary Fig. 4A–D). To investigate the robustness of PIANOS further, we iteratively removed random proportions of genes, reducing the input gene pool to 10%. Intriguingly, even with a 50% reduction in input genes, both C-index and D-index exhibited only a marginal decrease, maintaining a C-index of 0.65 and a D-index of 2 across all cohorts (Fig. 3I, J). To demonstrate the changes in model input with varying proportions of gene deletion, we displayed the alterations in ssGSEA scores when gene deletion rates reached 50 and 10%. Although the ssGSEA scores showed some variations, the relative ranking among different pathway pairs within individual samples remained largely consistent (Supplementary Fig. 5C). A comparative analysis of 105 previously published CRC prognostic models[23] demonstrated superior performance of PIANOS, as it consistently outperformed comparison models in all evaluated cases (Fig. 3K). This underscores the exceptional predictive accuracy and clinical relevance of PIANOS in stratifying patients with CRC and reinforces its potential as a significant advancement in CRC prognostic assessment.

### Multi-omics landscape in different PIANOS groups

To elucidate PIANOS's prognostic capabilities, we performed a model decomposition analysis revealing that most pathways identified by PIANOS are associated with cell proliferation and immune-related processes (Supplementary Fig. 5A). To identify the most critical transcriptomic signals within PIANOS, we ranked the genes from these pathways based on their frequency of occurrence (Supplementary Fig. 5B). Notably, genes such as DKK1, GZMB, THBS1, and CCL5—recognized for their key contributions to colorectal cancer prognosis—have been incorporated into several existing prognostic models[24–31]. We then examined mutation data from COCC, TCGA, and ACICAM. Fig. 4A and Supplementary Fig. 6A, B display the top 15 genes with the highest mutation frequencies in high- and low-risk groups, along with their mutation rates. To evaluate the model's stability, we combined mutation frequencies from these cohorts and identified commonalities, finding significant differences in MUC16 and FSIP2 mutation

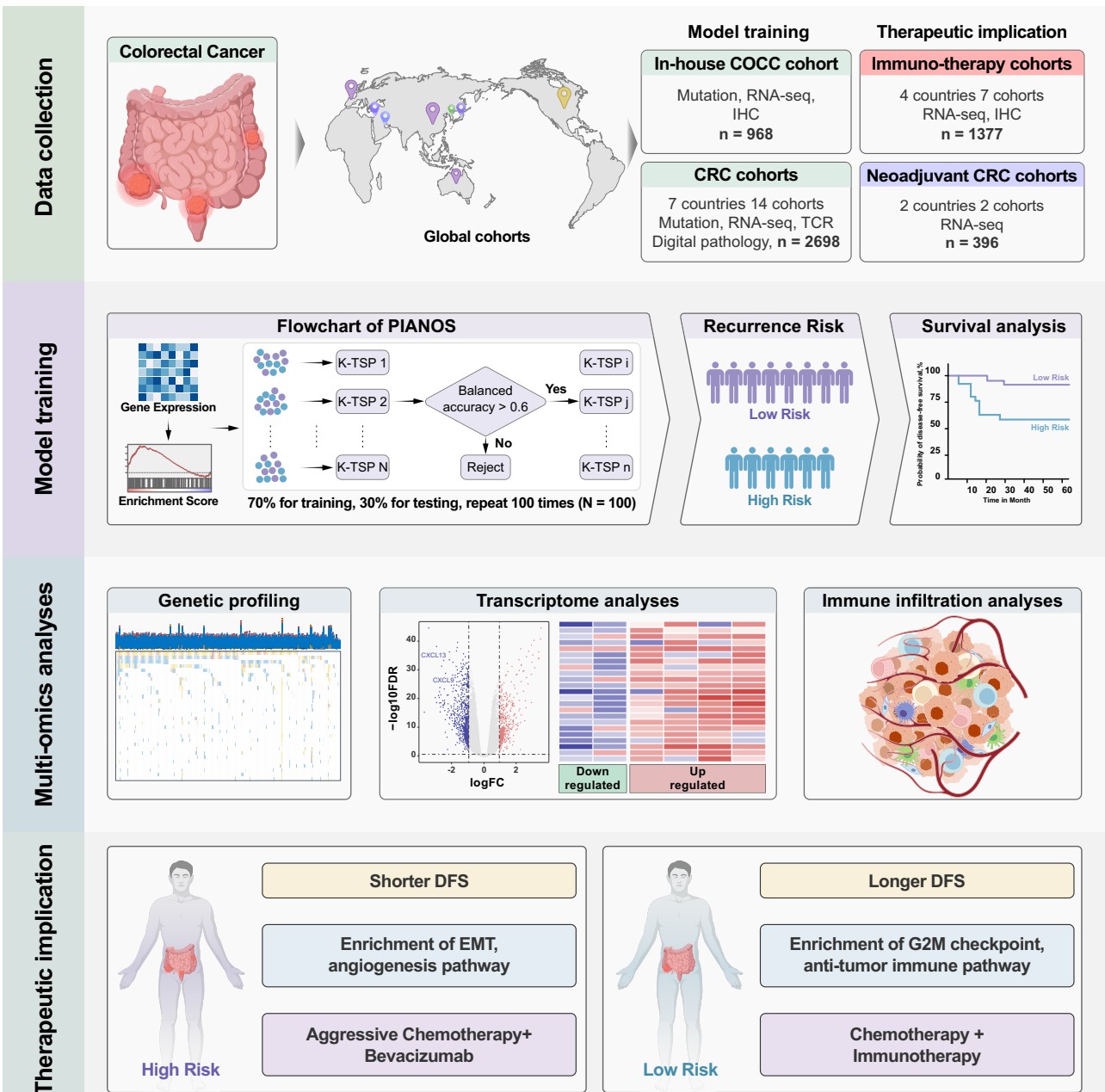

**Fig. 1 | Overview of the entire research process.** The study framework is organized into four main stages. First, Data collection: gene expression profiles were collected from a comprehensive set of 24 cohorts globally, totaling over 5000 patients. These were categorized for specific analyses. Second, Model training: This panel illustrates the flowchart for our model. Gene expression data from individual samples are first transformed into pathway enrichment scores. The k-top scoring pairs algorithm is then employed within the resampling process (100 iterations) to build a multi-classifier model. Our model stratifies colorectal cancer patients into high- and low-risk prognostic groups. Third, multi-omics analyses: To uncover the biological underpinnings of the risk stratification, we conducted integrated analyses of genetic profiles, transcriptomes, and the tumor immune microenvironment. Fourth, Therapeutic implication: Based on the distinct biological underpinnings of the risk groups, we showed the potential treatment strategies.

frequencies between high- and low-risk groups across all cohorts. Notably, these genes exhibited higher mutation frequencies in low-risk group (Fig. 4B). We computed the mutation status of high-frequency mutated genes exhibiting differential mutation frequencies in the COCC cohort, including APC, RYR2, SMAD4, TRCF7L2, TP53, FAT4, and MUC16 (Supplementary Fig. 6D). Furthermore, we performed differential gene expression analysis between high- and low-risk groups in major cohorts. Besides, we conducted survival analysis on differential genes in each cohort. The intersection of survival-contributing genes across four cohorts identified CXCL13 and CXCL9

(Fig. 4C and Supplementary Fig. 6C). Correlation between consensus molecular subtypes (CMS) and PIANOS risk groups is shown in Fig. 4D, Supplementary Figs. 6E, 7B, and Supplementary Data 15–17. Notably, CMS1 and CMS2 were predominantly associated with the low-risk group, whereas the CMS4 subtype was prevalent in the high-risk group. Analysis of HALLMARK pathways showed that IFN-α and IFN-γ response pathways were enriched in the low-risk group. Conversely, EMT (Epithelial-mesenchymal transition) and TGF-β signaling pathways were enriched in the high-risk group, which also showed significant activation of angiogenesis pathways (Fig. 4E and

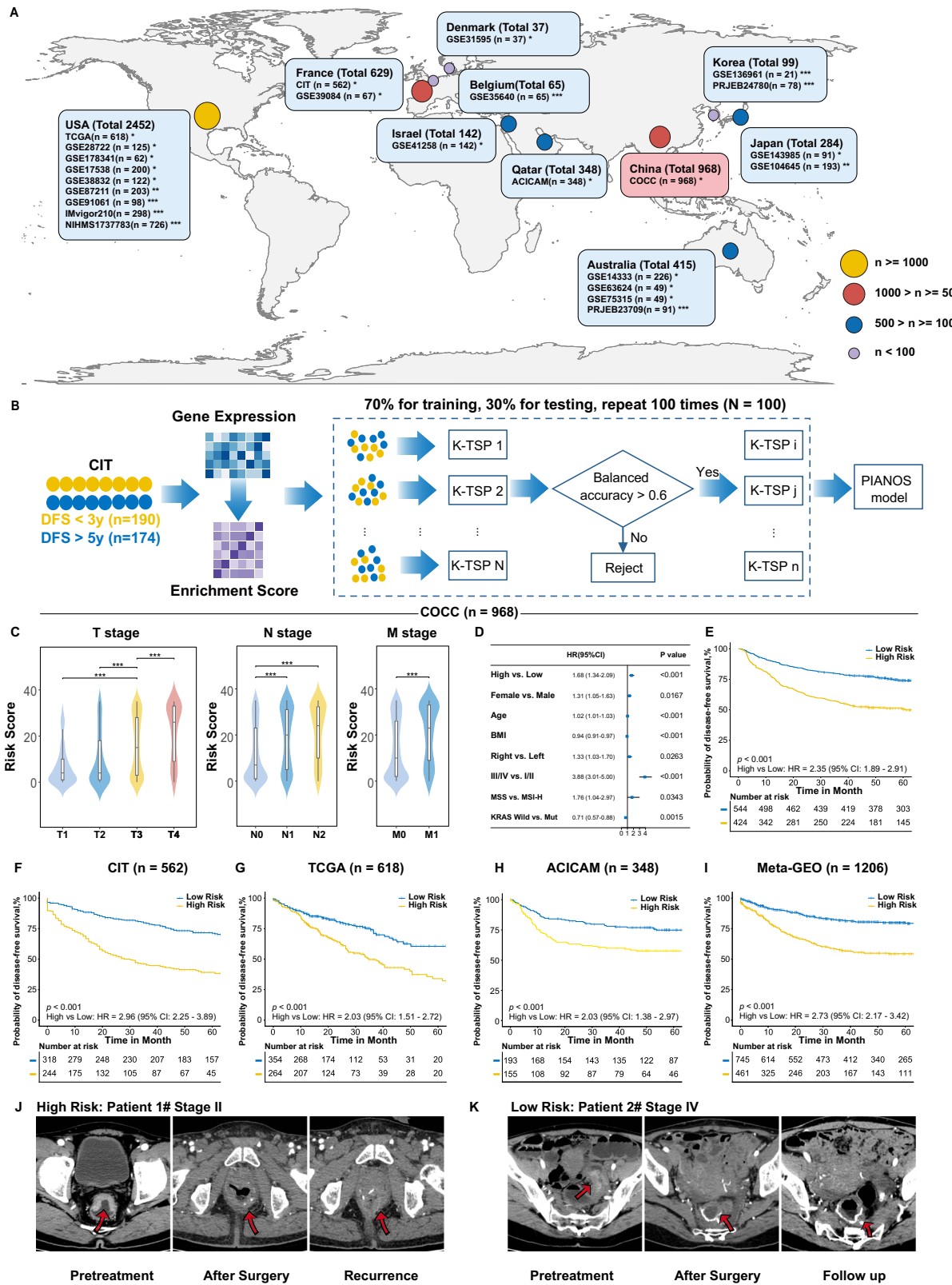

Supplementary Fig. 6F). To validate these findings, we evaluated ssGSEA scores for five treatment-relevant pathways across 10 GEO cohorts. This confirmed that TGF-β pathways and angiogenesis pathways were more active in the high-risk group (Supplementary Fig. 7A and Supplementary Data 18). Notably, G2M checkpoint pathway activation in the low-risk group was observed in all major cohorts except TCGA.

## Patients with CRC and low PIANOS scores were more sensitive to chemotherapy

Our analysis revealed a significant upregulation of G2M checkpoint pathway, a key target of several chemotherapeutic drugs[32], in low-risk group (Fig. 5A). To investigate chemotherapy sensitivity across risk groups, we searched Cancer Genome Project database and found that 5-fluorouracil, irinotecan, and oxaliplatin exhibited lower IC50 in low-

**Fig. 2 | Construction and evaluation of the prognostic integration analysis system (PIANOS) prognostic signature for colorectal cancer (CRC). A** Global distribution of the cohort included in this study. The in-house Clinical Omics Study of Colorectal Cancer in China (COCC) cohort is highlighted. *CRC validation cohorts; **CRC neoadjuvant therapy validation cohorts; ***Immunotherapy validation cohorts. **B** Schematic representation of the methodological workflow. **C** Violin plot of PIANOS risk scores versus TNM stage in COCC (n = 968). Intergroup comparisons (p values): T1 vs T3 (1.49 × 10⁻⁶), T2 vs T3 (2.45 × 10⁻⁸), T3 vs T4 (2.73 × 10⁻⁶), N0 vs N1 (2.39 × 10⁻¹²), N1 vs N2 (0.0359), M0 vs M1 (3.19 × 10⁻¹³). Box plots: median (center), 25th/75th percentiles (box), whiskers (1.5xIQR, outliers not shown). P values from the two-sided Wilcoxon rank-sum test. *p < 0.05, **p < 0.01, ***p < 0.001. **D** Forest plot of multivariable Cox regression analysis in COCC (n = 968), comparing PIANOS high- vs low-risk groups. Blue squares: hazard ratios (HRs); horizontal lines: 95% CI. P values from two-sided likelihood ratio test; exact values in Supplementary Data 9. **E–I** Kaplan–Meier curves for disease-free survival (DFS) by PIANOS group in COCC (p = 6.31 × 10⁻¹⁶), CIT (p = 1.58 × 10⁻¹⁹), TCGA (p = 4.87 × 10⁻⁶), ACICAM (p = 2.15 × 10⁻⁴), and meta-GEO (p = 5.95 × 10⁻²⁰) cohorts. P values from a two-sided log-rank test. **J, K** Representative CT scans of CRC patients before and after treatment. Source data are provided as a Source Data file.

risk group, indicating enhanced drug sensitivity (Fig. 5B). For 282 patients in COCC underwent Ki67 detection, immunohistochemistry (IHC) showed higher Ki67 expression in low-risk group (Fig. 5C) and a significant disparity in Ki67 distribution between high- and low-risk groups (Fig. 5D), with a positive correlation between Ki67 expression and G2M checkpoint activity (R = 0.26, p < 0.001, Fig. 5E). Stratifying patients into low (≤30%) and high (>30%) Ki67 groups, Kaplan–Meier curves indicated significantly better DFS for high Ki67 group patients receiving chemotherapy compared to untreated patients (HR = 0.40, p < 0.01, Fig. 5G). No significant DFS difference was noted in low Ki67 group based on chemotherapy administration (Fig. 5F). Chemotherapy benefits varied, with a 53% DFS improvement in the low-risk group versus 41% in the high-risk group (Fig. 5H, I). Data from GSE87211 (n = 203) showed a negative correlation between PIANOS scores and chemotherapy-induced regression rates, quantifying chemotherapy sensitivity based on pre- and post-treatment tumor invasion depth (Fig. 5J). In cohorts from GSE104645 (n = 193), the chemotherapy response rate was 69% in the low-risk group and 48% in the high-risk group (Fig. 5K). CT imaging (Fig. 5L) depicted post-chemotherapy outcomes for different risk groups, showing significantly prolonged DFS in low-risk patients receiving chemotherapy, whereas high-risk and untreated low-risk patients showed early recurrence. Baseline information of chemotherapy cohorts was presented in Supplementary Data 19–24.

## Metastatic CRC patients with high PIANOS scores exhibit higher angiogenesis activation features

Enrichment analysis revealed an enhanced angiogenesis pathway in the high-risk group (Fig. 4E), indicating that bevacizumab, a widely utilized monoclonal antibody in CRC therapy, might have potential clinical utility in this subgroup[33]. We confirmed disparities in HALLMARK angiogenesis and WP-angiogenesis gene sets across PIANOS risk groups in major cohorts (Fig. 6A, B). Subsequently, quantification of vascular endothelial cell abundance using xCell revealed a significant increase in the high-risk group (Fig. 6C). Moreover, VEGFA, a direct target of bevacizumab, showed elevated expression in the high-risk group (Fig. 6D). Using single-cell data from GSE178341[34], we performed cellular clustering and tSNE dimensionality reduction, focusing on stromal cells within epithelial populations (Fig. 6E–G). Patients with high-risk had a significantly larger proportion of VEGFA-positive stromal cells (Fig. 6H, I). We collected data on stage IV patients for analysis, including 30 patients who received bevacizumab treatment in the COCC cohort (baseline characteristics are presented in Supplementary Data 25–28). Due to the limited cohort size, we employed propensity score matching (PSM) to balance baseline differences between treated and untreated patients. Survival analysis of stage IV recurrent patients undergoing bevacizumab therapy in COCC, stratified by median VEGFA expression levels, revealed no OS difference in the low VEGFA group between treated and untreated patients (Fig. 6J). In contrast, in the high-VEGFA group, bevacizumab-treated patients demonstrated significantly improved OS compared to untreated patients (Fig. 6K, HR = 0.42, p = 0.047, 95% CI = 0.17–1.01). Applying similar analysis to PIANOS scores, no significant OS difference was observed in the low-risk group between treatment modalities (Fig. 6L), whereas the high-

risk group treated with bevacizumab showed significantly enhanced OS (Fig. 6M, HR = 0.43, p = 0.019, 95% CI = 0.21–0.89). Figure 6N, O shows CT images of typical high-risk group patients with or without bevacizumab treatment, respectively.

## Tumor immune microenvironment in different PIANOS groups

Initial analysis of immune response pathways, including IFN-α, T cell receptor signaling, and PD1 mediated immunity, demonstrated significant enrichment in the low-risk group (Fig. 7A and Supplementary Fig. 8A–C), suggesting a more active immune microenvironment. Representative H&E-stained images from TCGA highlighted these differences (Fig. 7B). Utilizing immune infiltration estimates by Saltz et al. derived from deep learning analysis of H&E-stained slides, we observed an abundance of lymphocytes, TILs (tumor-infiltrating lymphocytes), and macrophages in the low-risk group (Fig. 7C and Supplementary Data 29). Quantification of tumor microenvironment (TME) via xCell showed a substantial immunoscore elevation in the low-risk group, with increased infiltration of antitumor immune cells such as CD8 + T cells, natural killer T (NKT) cells, dendritic cells (DCs), and macrophages. Immune checkpoint and immunomodulatory genes confirmed an increased expression in the low-risk group (Fig. 7D and Supplementary Data 30). Additionally, intrinsic immune response markers, including tumor mutational burden (TMB), silent mutation rate, single-nucleotide variant (SNV) neoantigens, and insertion-deletion (indel) neoantigens, were significantly higher in the low-risk group (Fig. 7E). Extrinsic immune responses, characterized by leukocyte, lymphocyte, and TIL fractions and cytolytic activity (CYT) scores, were also elevated in the low-risk group (Fig. 7F). Low-risk tumors exhibited lower TIDE scores (Fig. 7G), indicating better responsiveness to immune checkpoint inhibitor (ICI) therapy. Comparing our findings with NIHMS958212's TCGA classification based on multi-omics data[35], we observed a strong correlation between the C2 subtype, known for high immune cell infiltration, and the low-risk group (Fig. 7H). Notably, the low-risk group had a higher prevalence of high microsatellite instability, known to favorably affect ICI therapy outcomes (Fig. 7I). Furthermore, we noted significant CXCL13 expression in TNKILC cells (T/natural killer [NK]/innate lymphoid cell [ILC]) (Fig. 7J). Dimensionality reduction and clustering of T cell subgroups showed a diminished proportion of TNKILC cells and CXCL13-positive cells among both CD8+ and CD4+ cells in the high-risk group (Fig. 7K–O). Analysis of TLS within the tumor milieu revealed a significantly lower number of TLS in the high-risk group than in the low-risk group (Fig. 7P, Q). In addition, TCR (T cell receptor) levels were higher in the low-risk group (Fig. 7R), and ICR (immunologic constant of rejection) scores from the original cohort supported similar conclusions (Fig. 7S) in ACICAM. This suggests that different PIANOS groups may have varying sensitivities to immunotherapy.

## Patients in the low-risk group may be more sensitive to immunotherapy

We explored the differences in ICI response across different PIANOS groups in several immunotherapy cohorts. In GSE35640, including 56 skin cutaneous melanoma (SKCM) patients undergoing MAGEA3 immunotherapy, the low-risk group had a higher response rate than

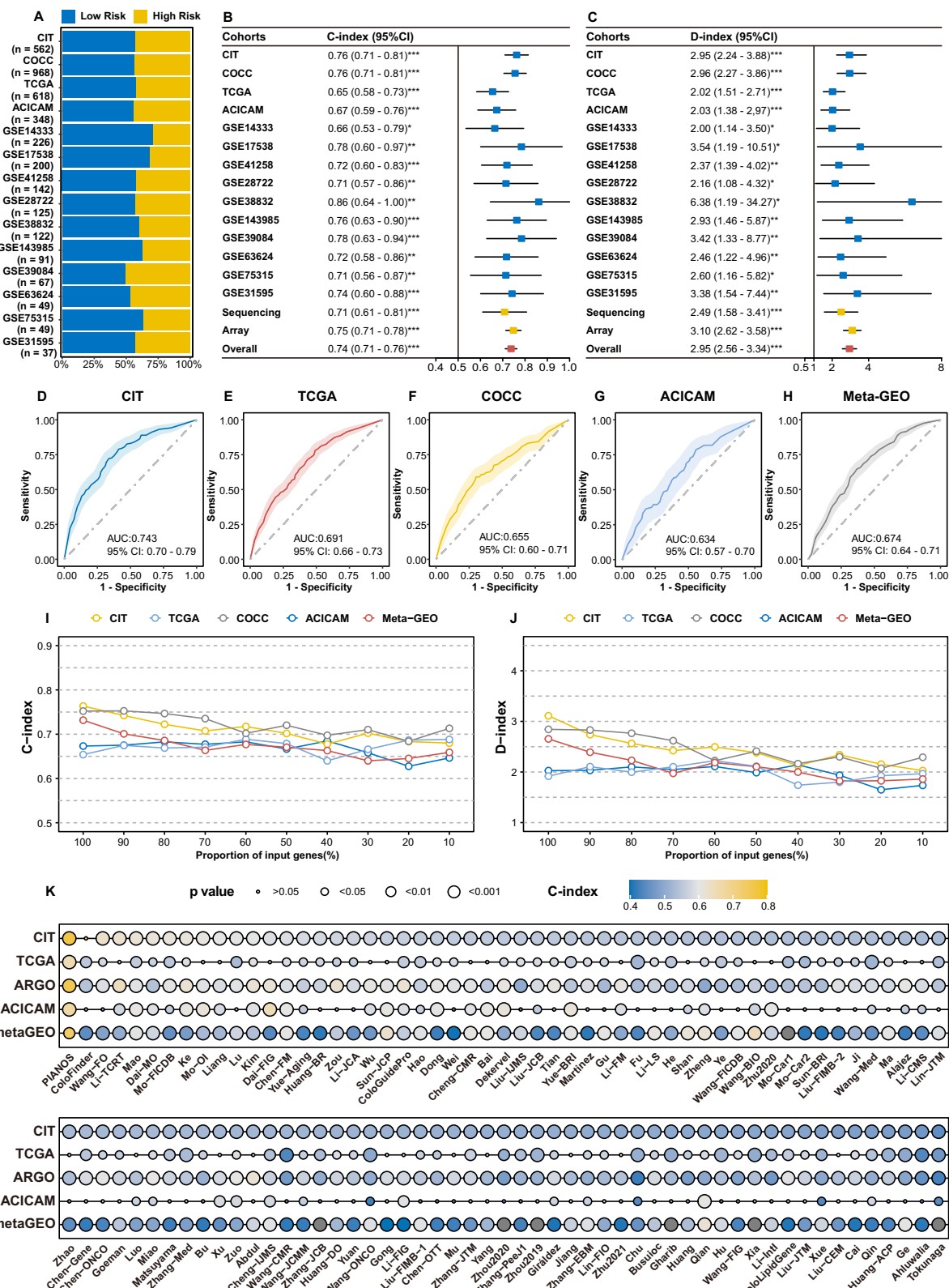

**Fig. 3 | Performance and robustness of the PIANOS model. A** Proportions of PIANOS high- and low-risk groups across all cohorts. **B**, **C** Forest plots of concordance indices (C-index) and robust hazard ratios (D-index) across cohorts, with meta-estimates for sequencing, array-based, and combined platforms. Cohorts and sample sizes (*n*): CIT (562), COCC (968), TCGA (618), ACICAM (348), GSE14333 (226), GSE17538 (200), GSE41258 (142), GSE28722 (125), GSE38832 (122), GSE143985 (91), GSE39084 (67), GSE63624 (49), GSE75315 (49), and GSE31595 (37). Squares: point estimate; horizontal lines: 95% CI. Significance for D-index via two-sided log-rank test;

significance for both indices vs. reference (D-index=1, C-index=0.5) inferred from 95% CIs. No multiple comparison adjustments. Exact *p* values in Supplementary Data 11, 12. *$p < 0.05$, **$p < 0.01$, ***$p < 0.001$. **D**–**H** Area under the ROC curve (AUC) for PIANOS predicting three-year recurrence in CIT, TCGA, COCC, ACICAM, and Meta-GEO cohorts. Solid line: ROC curve; shaded area: 95% CI. **I**, **J** Line charts illustrating C-index and D-index changes with input gene loss. **K** Bubble plot comparing C-index and *p* values of PIANOS versus 105 other models across all cohorts. *P* values from a two-sided permutation test. Source data are provided as a Source Data file.

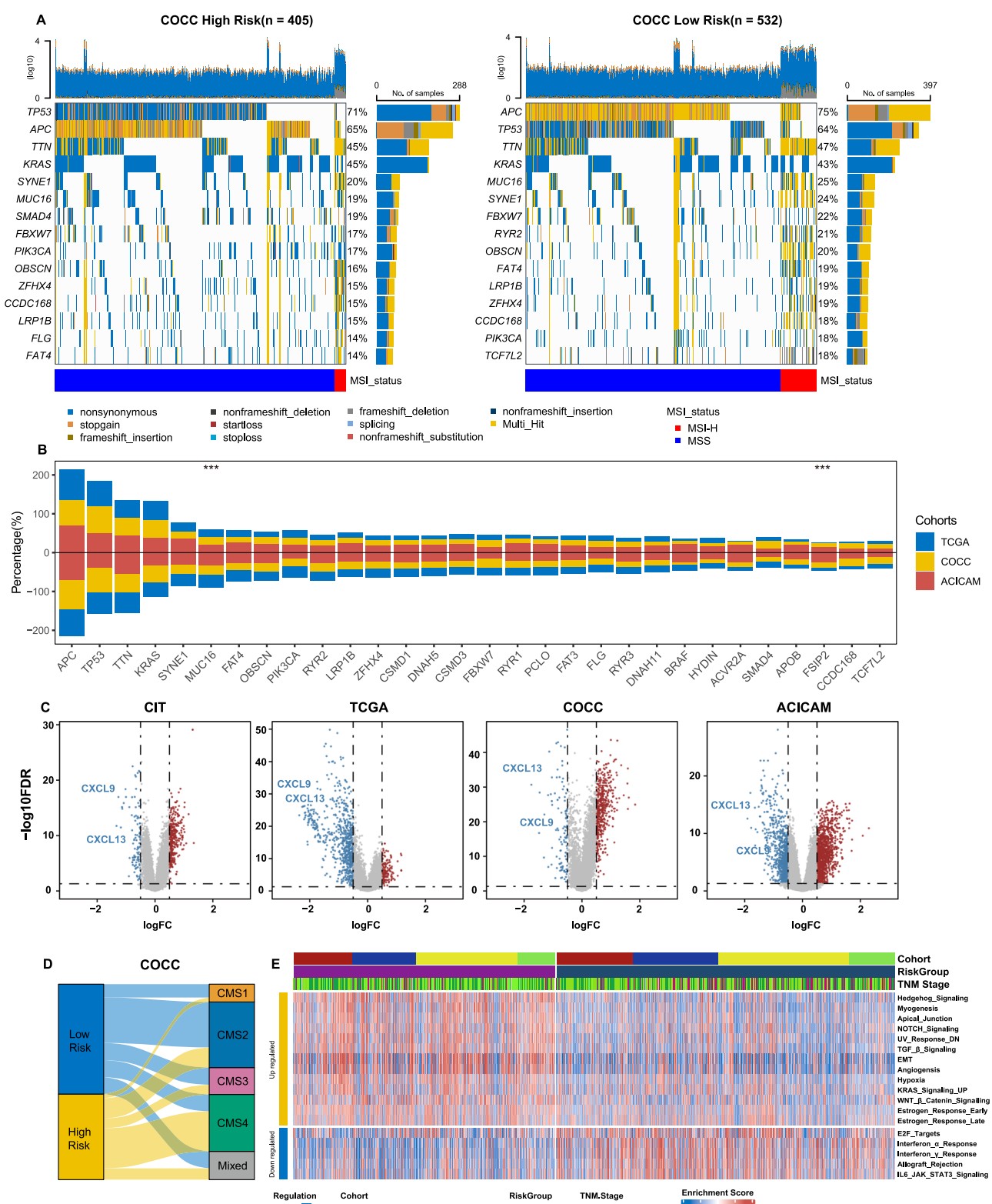

**Fig. 4 | Multi-omics landscape between PIANOS groups. A** Waterfall plot of the top 15 most frequently mutated genes by the PIANOS group in COCC. **B** Stacked bar plot of genes with the highest mutation frequency across three cohorts (COCC, TCGA, ACICAM). Genes with significantly different mutation frequencies (high- vs low-risk) across all three cohorts are marked (***). *P* values for FSIP2 and MUC16, respectively: COCC (0.020, 0.040), TCGA (0.007, 0.004), ACICAM (0.048, 0.040); two-sided Chi-square test. **C** Differential genes between high- and low-risk groups,

along with the intersection of all survival-related differential genes across all cohorts, resulting in CXCL13 and CXCL9. **D** Sankey diagram illustrating the relationship between PIANOS groups and consensus molecular subtypes (CMS). **E** Heatmap of enriched differential HALLMARK pathways by PIANOS group across CIT, TCGA, ACICAM, and COCC cohorts. Source data are provided as a Source Data file.

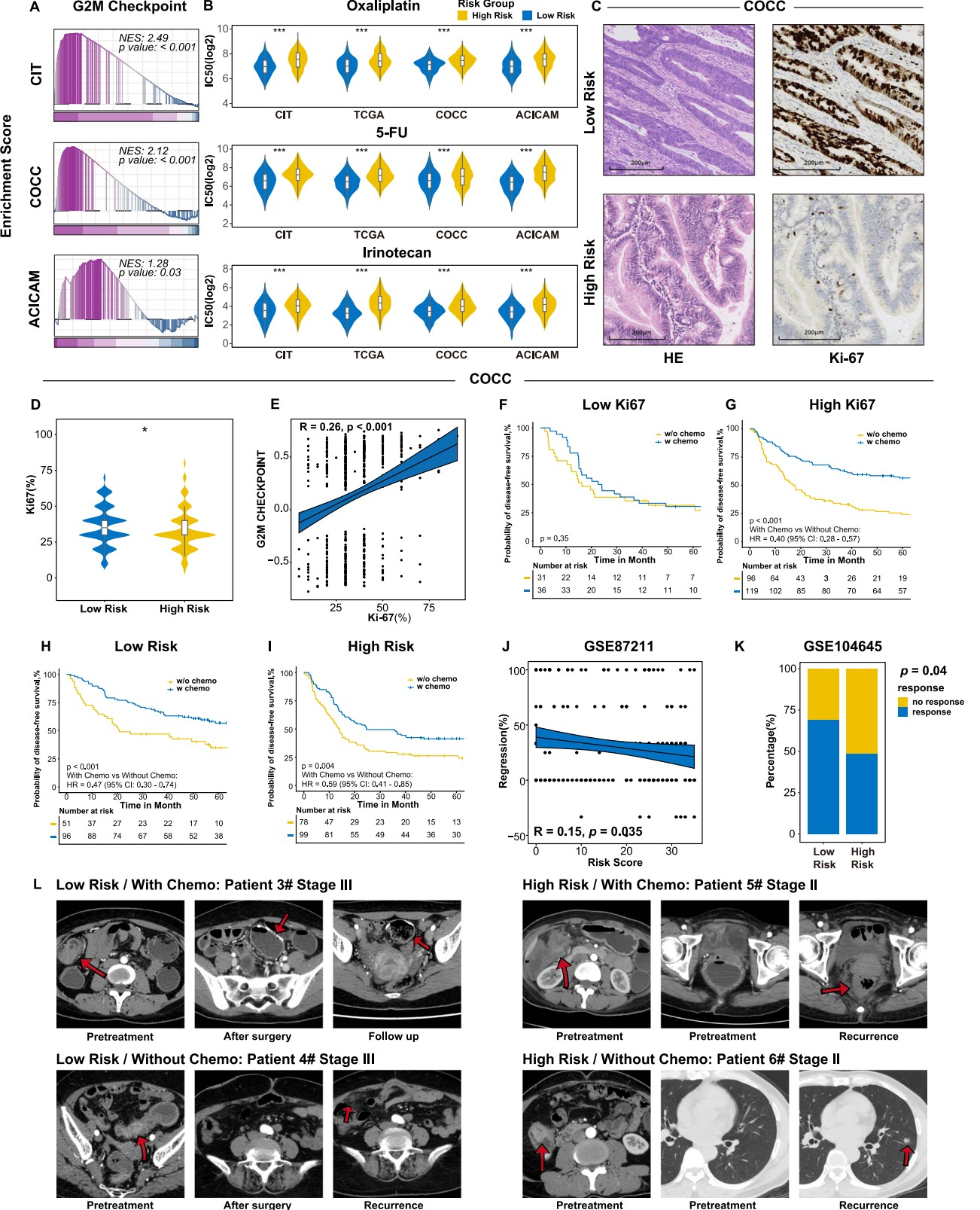

the high-risk group (59 vs. 19%, Fig. 8A). Similarly, GSE91061, with 98 SKCM patients treated with PD1 immunotherapy, showed a higher proportion of objective response rate (ORR) in the low-risk group than in the high-risk group (27 vs. 16%,). PSM further revealed superior survival outcomes in low-risk group (HR = 2.75, $p$ = 0.04, Fig. 8B). In a cohort of 21 patients with non-small cell lung cancer (NSCLC) (GSE136961) receiving PD1 blockade, PIANOS score achieved an AUC of

0.778 for predicting immunotherapy responses (Fig. 8C). IMvigor210, encompassing 298 urothelial cancer (UC) patients treated with PD-L1 inhibitors, showed higher response rates (28 vs. 17%) and enhanced survival in low-risk group compared to high-risk group (HR = 1.52, $p$ = 0.03, Fig. 8D). In PRJEB23709, which included 78 renal cell carcinoma (RCC) patients undergoing PD1 immunotherapy, low-risk group also had a higher response rate (67 vs. 41%) and better survival

**Fig. 5 | CRC patients with a low PIANOS score were more sensitive to chemotherapy. A** Gene set enrichment analysis (GSEA) plot of the G2M checkpoint pathway (low- vs high-risk). Datasets: CIT($p < 1 \times 10^{-10}$), ACICAM($p = 0.031$), COCC($p < 1 \times 10^{-10}$). Two-sided permutation test; $p$ values Benjamini–Hochberg adjusted. **B** 50% inhibitory concentration (IC50) values for oxaliplatin, irinotecan, and fluorouracil (5-FU) by PIANOS risk group. Datasets and drugs ($n$, $p$ values provided for irinotecan, oxaliplatin, 5-FU respectively): CIT ($n = 562$; $1.81 \times 10^{-7}$, $1.87 \times 10^{-13}$, $3.45 \times 10^{-14}$), COCC ($n = 968$; $4.06 \times 10^{-22}$, $1.79 \times 10^{-18}$, $1.99 \times 10^{-8}$), TCGA ($n = 618$; $3.24 \times 10^{-31}$, $1.56 \times 10^{-12}$, $8.56 \times 10^{-18}$), ACICAM ($n = 348$; $1.90 \times 10^{-10}$, $1.19 \times 10^{-9}$, $1.11 \times 10^{-17}$). **C** Representative hematoxylin and eosin (H&E) staining and Ki67 immunostaining of tumor tissues. Image is illustrative due to specimen availability; see Fig. 5D for quantitative analysis. **D** Violin plots of Ki67(high- vs low-risk) in COCC ($n = 536$, $p = 0.015$). **E** Correlation between G2M checkpoint and Ki67 in COCC ($p = 1.4 \times 10^{-9}$). Plot shows data points (black circles), a linear regression line, and 95% CI (shaded area). **F, G** Kaplan–Meier curves for disease-free survival (DFS) by chemotherapy status in COCC patients stratified by Ki67 (high: $p = 1.68 \times 10^{-7}$; low: $p = 0.35$). **H, I** Kaplan–Meier curves for DFS by chemotherapy status in COCC patients stratified by PIANOS risk (high: $p = 8.20 \times 10^{-4}$; low: $p = 0.001$). **J** Correlation between PIANOS risk scores and regression rates in GSE87211. Plot shows data points (black circles), a linear regression line, and 95% CI (shaded area). **K** Proportions of chemotherapy sensitivity by PIANOS risk group in GSE104645 ($p$ value from two-sided Chi-square test). **L** Representative CT images showing chemotherapy impact by PIANOS risk group. Unless otherwise stated, $p$-values calculated using two-sided Wilcoxon rank-sum test (**B, D**) or log-rank test (**F–I**). For regression plots (**E, J**), $p$ values from the two-sided Wald test. Box plots: median (center), 25th/75th percentiles (box), whiskers (1.5xIQR, outliers not shown). $*p < 0.05$, $**p < 0.01$, $***p < 0.001$ (**D**). Source data are provided as a Source Data file.

outcomes (HR = 2.21, $p = 0.02$, Fig. 8E). Analysis of 78 RCC patients received PD1 immunotherapy from PRJEB25780 corroborated higher immunotherapy response rate in low-risk group (39%) than high-risk group (14%) (Fig. 8F). An evaluation of 726 RCC patients from NIHMS1737783 treated with PD-L1 immunotherapy showed low-risk group had a higher response rate (80%) than high-risk group (61%). IHC data from this cohort validated significant enrichment of CD8+ and PD-L1+ cells, and elevated CD274 and PD-L1 expression levels in the low-risk group (Fig. 8G).

## Discussion

Accurate identification of patients at high risk for CRC remains a formidable challenge in clinical diagnostics and treatment. Achieving accuracy in prognosis, a crucial aspect of personalized medicine, is hindered by tumor heterogeneity and limitations of current algorithms[20,36]. These factors complicate the use of genomic expression-based prognostic models in clinical settings. PIANOS, developed using the pathway k-TSP algorithm, resampling techniques, and model integration, marks a pioneering effort to establish a stable prognostic prediction system applicable across diverse cohorts. Tested across 15 cohorts with 3666 patients, PIANOS demonstrated exceptional predictive performance and may indicate differences in treatment sensitivities among patient subgroups. This underscores its potential ability to enhance stratified patient management in CRC.

In CRC prognosis, PIANOS outperformed 105 existing molecular classification models across nearly all cohorts. Compared to traditional methods that require predefined cohort construction as a baseline reference—essentially focusing on inter-sample differences—PIANOS employs an innovative approach to identify intra-sample variability, effectively bypassing the issue of cohort standardization. This significantly enhances model performance and versatility, enabling PIANOS to be applied to genomic sequencing data from any source[37,38]. Additionally, by shifting from solely relying on gene expression to incorporating pathway enrichment as the primary input, PIANOS effectively manages scenarios with missing genes[39]. These features significantly enhance its clinical utility, offering a versatile and robust prognostic tool that addresses challenges such as tumor heterogeneity and incomplete data.

PIANOS aligns closely with the TNM system in recognizing tumor heterogeneity and provides additional insights[40]. To elucidate prognostic disparities identified by PIANOS, we investigated variations in pathway enrichment. Remarkably, patients classified in the high-risk group exhibited significant upregulation of the EMT pathway, indicating tumor invasiveness and poorer prognosis[41]. Existing research indicates that patients with CMS4 subtype exhibit low immune activity, high invasiveness and metastatic potential, prominent stromal component, and gene expression features related to tumor growth and dissemination, such as angiogenesis and TGF-β signaling pathways. These characteristics contribute to a poorer prognosis and are

consistent with traits observed in our high-risk group[42]. Conversely, activation of antitumor immune pathways and G2M checkpoint in the low-risk group highlights distinct phenotypic characteristics and potential therapeutic responses. Mutational analysis revealed a higher prevalence of MUC16 and FSIP2 mutations in the low-risk group, which is consistent with previous studies suggesting enhanced responsiveness to immunotherapy in solid tumors harboring MUC16 mutations[43,44]. In the COCC cohort, mutation analysis demonstrated that the high-risk group exhibited an elevated mutation frequency of SMAD4 and TP53. Previous studies have indicated that these features are associated with enhanced EMT and angiogenesis[45,46], whereas wild-type TP53 may confer a more robust immune phenotype[47]. Moreover, mutations in TCF7L2, FAT4, MUC16, and RYR2 have been correlated with heightened immune activity and improved prognosis in CRC and other malignancies[48–51]. Through integration of differential gene expression and survival analysis, we identified that the low-risk group exhibited elevated expression levels of CXCL9 and CXCL13. This observation indicates enhanced T cell activity and macrophage polarization within low-risk cohort[52–54]. Collectively, these observations suggest that low-risk patients may exhibit greater sensitivity to immunotherapy, while high-risk patients may demonstrate increased responsiveness to angiogenesis-targeting treatments.

Adjuvant chemotherapy is crucial for reducing recurrence risk and improving patient prognosis in CRC managing[55]. Current clinical practices primarily rely on pathological assessments and staging criteria[56]. However, tumor complexity and heterogeneity pose substantial challenges to the precision of these criteria, potentially leading to over- or under-treatment[4]. Given the adverse effects of chemotherapy and the impact of tumor recurrence on patient survival, personalized treatment is essential. Common CRC chemotherapeutic agents (e.g., 5-fluorouracil, oxaliplatin, and irinotecan) target highly proliferative tumor cells, suggesting that tumor proliferative activity may indicate drug sensitivity. Previous studies have established a strong correlation between proliferation marker Ki67, which is commonly used in clinical settings, and chemotherapy sensitivity in patients with CRC[57]. Our analysis found elevated Ki67 levels in PIANOS low-risk patients, indicating increased tumor proliferation and greater chemotherapy sensitivity. This correlation was confirmed in COCC, where low-risk patients showed significant benefits from standard chemotherapy protocols. These insights suggest PIANOS could stratify patients in future chemotherapy clinical trials, optimizing treatment outcomes. Furthermore, consistent outcomes in two novel adjuvant chemotherapy cohorts indicate PIANOS' potential for guiding neoadjuvant treatment strategies. However, further clinical trials are needed to validate the robustness of PIANOS as a predictive tool, confirming its applicability and effectiveness in enhancing treatment precision and improving patient outcomes in CRC therapy.

Bevacizumab, a humanized anti-VEGFA monoclonal antibody, is a primary treatment for advanced metastatic CRC[33]. Despite its widespread use, a comprehensive decision-making framework for its

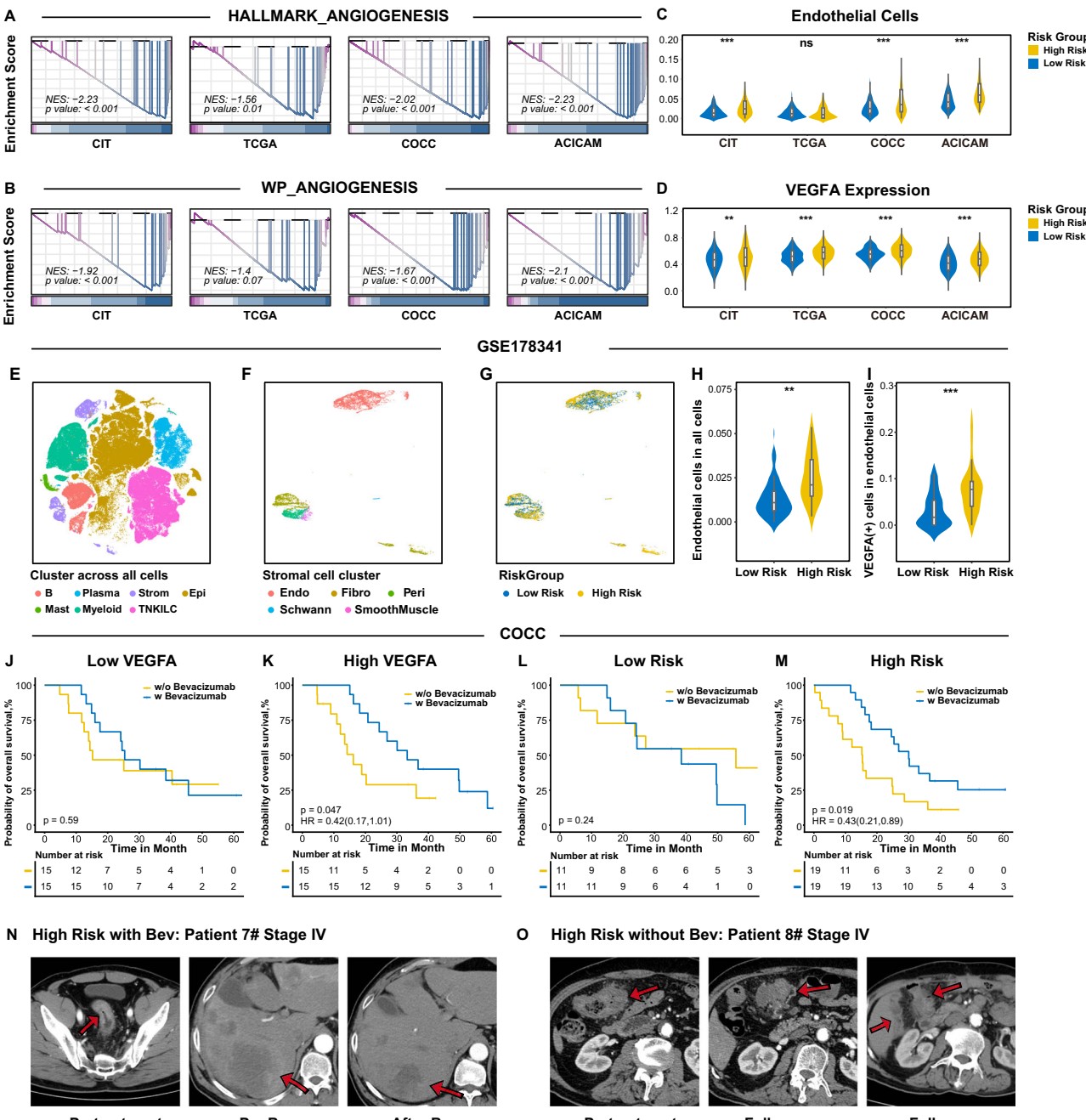

**Fig. 6 | Metastatic CRC patients with a high PIANOS score were more sensitive to bevacizumab.** **A**, **B** Gene set enrichment analysis (GSEA) plots of angiogenesis(CIT $p = 7.36 \times 10^{-9}$, TCGA $p = 0.01$, ACICAM $p = 1.93 \times 10^{-7}$, COCC $p = 5.92 \times 10^{-10}$) and WP_angiogeneis(CIT $p = 2.70 \times 10^{-4}$, TCGA $p = 0.07$, ACICAM p = $1.00 \times 10^{-5}$, COCC $p = 2.65 \times 10^{-3}$) pathway(low- vs high-risk). *P* values from a two-sided permutation test. **C**, **D** Violin plots of predicted endothelial cells (CIT $p = 1.92 \times 10^{-9}$, TCGA $p = 0.702$, ACICAM $p = 3.06 \times 10^{-7}$, COCC $p = 9.71 \times 10^{-11}$), and VEGFA expression(CIT $p = 7.71 \times 10^{-3}$, TCGA $p = 1.11 \times 10^{-7}$, ACICAM $p = 1.06 \times 10^{-4}$, COCC $p = 1.17 \times 10^{-8}$) by PIANOS risk group. Datasets: CIT ($n = 562$), COCC ($n = 968$), TCGA ($n = 618$), ACICAM ($n = 348$). **E**–**G** t-distributed stochastic neighbor embedding (tSNE) plot showing cell clusters (all cells and stromal cells) in GSE178341. **H**, **I** Violin plots showing endothelial cells in all cells($p = 0.0013$) and VEGFA (+) cells

in endothelial cells ($p = 6.3 \times 10^{-4}$) by PIANOS risk group in GSE178341 ($n = 62$). **J**, **K** Kaplan–Meier curves for overall survival (OS) by bevacizumab status in COCC patients stratified by VEGFA expression (high and low). **L**, **M** Kaplan–Meier curves for OS by bevacizumab status in COCC patients stratified by PIANOS risk (high- and low-risk). **N**, **O** Representative CT images of high-risk patients before and after treatment. **N** Patient 7 (bevacizumab-treated) showing reduced liver metastases. **O** Patient 8 (bevacizumab-untreated) showing rapid progression. Unless otherwise stated, *P*-values for violin plots (**C**, **D**, **H**, **I**) were calculated using a two-sided Wilcoxon rank-sum test. *P* values for Kaplan–Meier curves (**J**–**M**) from two-sided log-rank test. Box plots: median (center), 25th/75th percentiles (box), whiskers (1.5xIQR, outliers not shown). \**p* < 0.05, \*\**p* < 0.01, \*\*\**p* < 0.001 (**C**, **D**, **H**, **I**). Source data are provided as a Source Data file.

application beyond clinical staging is lacking[58]. Adverse effects, including hypertension and bleeding risk, highlight the importance of identifying patients who would benefit most from bevacizumab. VEGFA expression level, the target of bevacizumab, is highly correlated with treatment benefits. However, defining cutoff values for VEGFA

expression remains inconsistent across different populations and assays. Our study identified that high-risk patients exhibited elevated VEGFA expression, enhanced angiogenic pathway activity, and a greater abundance of endothelial cells expressing high-VEGFA levels–features that collectively indicate robust angiogenic activation.

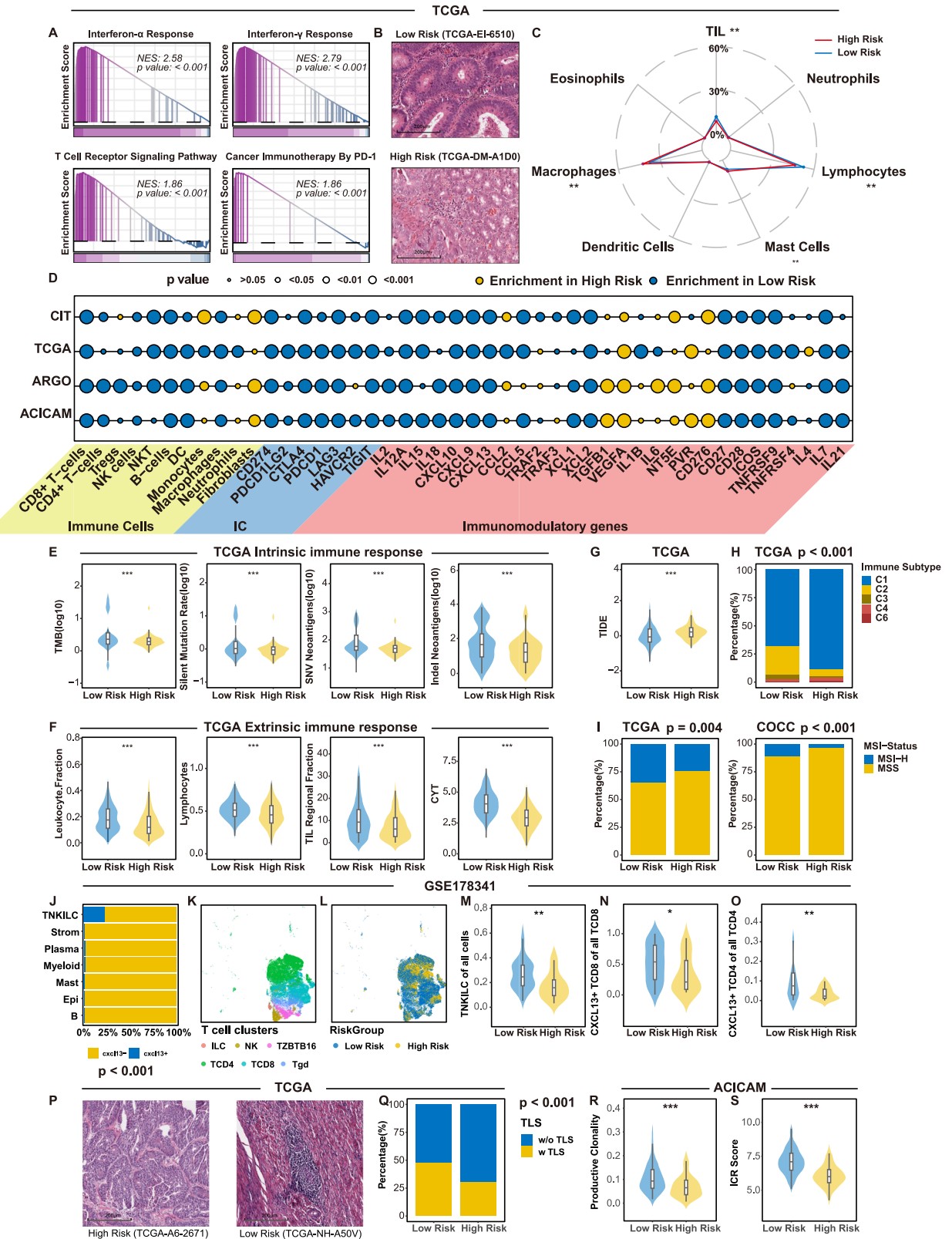

Encouragingly, our analysis of a small cohort of stage IV patients demonstrated a promising trend toward improved overall survival in the high-risk group receiving bevacizumab compared to those not treated. Although these preliminary findings derived using PSM support the potential predictive value of the PIANOS model, larger and more rigorously designed clinical studies are essential to confirm these observations and to further establish its clinical utility.

Our analysis found that low-risk patients displayed higher antitumor immune characteristics, suggesting potential sensitivity to immunotherapy. Comparing immune microenvironments between risk groups, we found that low-risk patients possess a hotter (more active) immune microenvironment characterized by higher expression of immune regulatory[59] and checkpoint genes[60], activation of immune pathways, and enrichment of immune cells[61,62]. Both

**Fig. 7 | Tumor immune microenvironment in different PIANOS groups. A** GSEA plots of interferon-α response($p < 1 \times 10^{-10}$), interferon-γ response ($p < 1 \times 10^{-10}$), T cell receptor signaling ($p = 4.81 \times 10^{-5}$) and cancer immunotherapy by PD1 ($p = 2.59 \times 10^{-4}$) pathway (low- vs high-risk) in TCGA. *P* values from a two-sided permutation test. **B** Representative HE staining images of tumor tissues in TCGA. The image is illustrative. See Fig. 7C for quantitative analysis. **C** Radar plot of immune cell counts (deep learning-predicted) by PIANOS risk group. **D** Bubble plot showing immune cell abundance and immunomodulatory/checkpoint gene expression(low- vs high-risk). **E** Comparison of the tumor mutational burden (TMB) ($n = 543$, $p = 4.3 \times 10^{-4}$), silent mutation ($n = 594$, $p = 2.5 \times 10^{-4}$), single nucleotide variants(SNV) neoantigens ($n = 594$, $p = 6.1 \times 10^{-7}$), and indel neoantigens ($n = 594$, $p = 1.2 \times 10^{-4}$) by PIANOS risk group in TCGA. **F** Comparison of leukocyte fraction ($n = 594$, $p = 9.8 \times 10^{-9}$), lymphocytes ($n = 594$, $p = 1.0 \times 10^{-6}$), tumor-infiltrating lymphocyte (TIL) region fraction ($n = 594$, $p = 4.6 \times 10^{-5}$), and cytolytic activity(CYT) score ($n = 594$, $p < 1.0 \times 10^{-15}$) by PIANOS risk group in TCGA. **G** Predicted TIDE score by PIANOS risk group in TCGA ($n = 618$, $p = 3.8 \times 10^{-7}$). **H** Immune microenvironment types by PIANOS risk group in TCGA ($p = 3.4 \times 10^{-9}$). **I** MSI-H proportions by PIANOS risk group in TCGA ($p = 0.004$) and COCC ($p = 6.17 \times 10^{-6}$). **J** CXCL13 expression across cell clusters ($p < 1 \times 10^{-15}$). **K, L** t-distributed stochastic neighbor embedding (tSNE) plot of T cell clusters in GSE178341. **M–O** Violin plots showing TNKILC (T cells, natural killer cells, innate lymphoid cell types) of all cells ($p = 0.006$), and CXCL13+ cells of all TCD8 ($p = 0.033$) and TCD4 ($p = 0.0051$) cells by PIANOS risk group in GSE178341 ($n = 62$). **P** Representative H&E staining of tertiary lymphoid structure(TLS) in TCGA. Illustrative example; see Fig. 7Q for quantitative analysis. **Q** TLS differences by PIANOS risk group ($p = 1.05 \times 10^{-4}$). **R, S** Differences of productive clonality ($n = 114$, $p = 8.09 \times 10^{-4}$) and ICR score ($n = 348$, $p = 6.49 \times 10^{-22}$) by PIANOS risk group in ACICAM. Unless otherwise stated: *p* values for GSEA(A) from two-sided permutation test. *P* values for comparisons/differences in (**C–G, M–O, R, S**) from two-sided Wilcoxon rank-sum test. *P* values for proportions/differences in (**H, I, Q, J**) from a two-sided Chi-square test. Box plots (**E–G, M–O, R, S**): median(center), 25th/75th percentiles(box), whiskers (1.5xIQR, outliers not shown). \**p* < 0.05, \*\**p* < 0.01, \*\*\**p* < 0.001 (**E–G, M–O, R, S**). Source data are provided as a Source Data file.

endogenous and exogenous immune responses, indicated by TMB[63] and CYT[64], respectively, supported these findings. Previous research utilizing data from TCGA categorized CRC into distinct immune subtypes. The C2 subtype, characterized by elevated CD8 signaling, high TCR diversity, and increased proliferative activity, indicative of a highly active immune response, was predominantly found in patients classified as low-risk by our model. MSI-H status, a marker of immunotherapy sensitivity in CRC[65], was more prevalent in the low-risk group, suggesting a favorable response to immunotherapy. Moreover, low-risk patients displayed a higher proportion of CXCL13 + CD4+ and CXCL13 + CD8 + T cells, indicating a more active tumor immune microenvironment and enhanced responsiveness to immunotherapy[66]. The presence of TLSs within tumors, associated with improved response to immunotherapy[67], was more common in low-risk patients, as confirmed by TCGA pathology slides, suggesting a robust antitumor immune capability. Additionally, biomarkers related to ICI response, such as TCR and ICR scores, were elevated in the low-risk group. These findings, validated in seven external immunotherapy cohorts, demonstrate that low-risk patients exhibit better treatment responses and prognosis. This evidence suggests that PIANOS could be valuable in guiding patient selection for immunotherapy, potentially improving clinical outcomes by identifying those most likely to benefit from such treatments.

As described in the Methods section, given our aim to develop a broadly applicable personalized risk assessment model for colorectal cancer, sex-related analyses were not implemented. Although our model effectively personalizes risk classification and provides some potential treatment inclinations, our study is retrospective, and the bevacizumab-related analysis is based on a limited small-scale cohort. Therefore, prospective studies in larger cohorts are required to verify the predictive performance of the model and the drug sensitivity of the stratified groups. Additionally, Further optimization may be needed to streamline throughput and reduce model complexity.

In conclusion, we found that PIANOS can serve as a robust molecular subtyping model to guide patient prognostic evaluation and suggest potential sensitivities to various treatment options for CRC. Patients in low-risk group may benefit from chemotherapy and immunotherapy regimens, whereas those in high-risk group exhibit activated angiogenic features and may be more sensitive to bevacizumab combined with chemotherapy. These results indicate that PIANOS not only serves as a complementary tool for TNM staging system but also enhances application of precise treatment strategies.

## Methods
This study was conducted in accordance with all relevant ethical regulations and was approved by the Medical Ethics Committee of the Sixth Affiliated Hospital of Sun Yat-sen University (Approval No. 2024ZSLYEC-408).

### Datasets
To construct and validate a cross-platform, robust single-sample risk stratification system, our research included cohorts from various countries and regions. We obtained gene expression profiles and follow-up data of 3666 patients with CRC[34,68–83] to validate PIANOS, 396 patients with CRC[84,85] to predict neoadjuvant therapy efficacy, including data from the Gene Expression Omnibus (GEO), The Cancer Genome Atlas (TCGA), and in-house Clinical Omics Study of Colorectal Cancer in China (COCC) CRC cohorts. GEO cohorts were downloaded from the GEO website (https://www.ncbi.nlm.nih.gov/geo/) and pre-processed using the GEOquery R package[86]. The French multicenter cohort (CIT) served as a training dataset while ten GEO cohorts were merged into the Meta-GEO validation cohort. Transcripts per million (TPM) data and H&E slides from COAD and READ TCGA cohorts were downloaded from (https://portal.gdc.cancer.gov/) and merged into the TCGA CRC validation cohort. The COCC CRC cohort for independent validation was the CRC subproject of the ICGC-ARGO project (https://www.icgc-argo.org/page/114/cgcc), and all cases were collected from the Sixth Affiliated Hospital of Sun Yat-sen University, Guangzhou, China. Written informed consent was obtained from all participants prior to sample collection. Participant sex was determined retrospectively from hospital medical records, based on the legal sex registered in each individual's national Resident Identity Card in China. All patients underwent curative surgical resection, and the diagnosis of colorectal cancer was rigorously confirmed by histopathological evaluation. Transcriptome data of 46 CRC cell lines were downloaded from the Cancer Cell Line Encyclopedia project (https://portals.broadinstitute.org/ccle) for drug sensitivity analysis. We collected seven immunotherapy cohorts with 1377 patients[87–93] to explore PIANOS' predictive value on immunotherapy response.

The primary endpoint of this study is disease-free survival (DFS), defined as the period of time after a patient has completed treatment or achieved complete remission, during which the patient remains free of any signs or symptoms of the disease. This includes time and status of recurrence or death. The bevacizumab-treated subgroup comprised metastatic colorectal cancer patients who underwent standard curative resection and, following tumor recurrence, received combination therapy with bevacizumab and chemotherapy. Accordingly, overall survival (OS), defined as duration from start of treatment until death from any cause or the last follow-up, was employed as the endpoint in our survival analysis. To reduce potential selection bias and improve result accuracy, we employed 1:1 propensity score matching (PSM) in stage IV patients based on variables including age, sex, and tumor location.

### Framework of PIANOS
PIANOS comprises three main components: pathway enrichment calculations, rank-based k-Top Scoring Pairs (k-TSP), and resampling. To more precisely identify characteristics distinguishing patients with

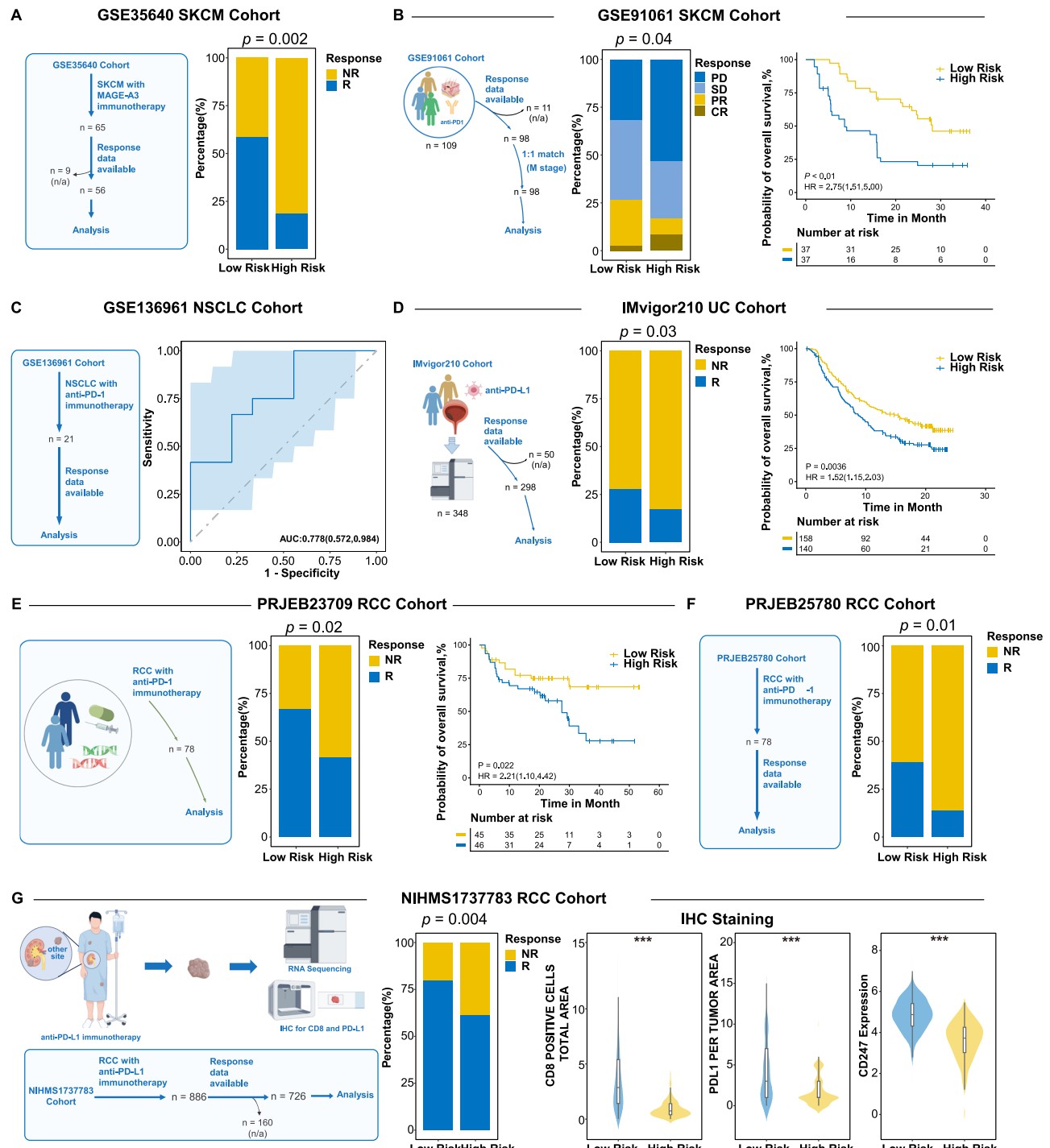

**Fig. 8 | Patients with a low PIANOS score can benefit more from immunotherapy. A** GSE35640 (SKCM): Immune checkpoint inhibitor (ICI) response proportions by PIANOS risk group. **B** GSE91061 (SKCM): ICI response proportions and Kaplan–Meier overall survival (OS) curves($p = 9.01 \times 10^{-4}$) by PIANOS risk group. **C** GSE136961 (NSCLC): Area under the ROC curve (AUC) for PIANOS predicting OS. Solid line: ROC curve; shaded area: 95% CI. **D** IMvigor210 (UC): ICI response proportions and Kaplan–Meier OS curves by PIANOS risk group. **E** PRJEB23709 (RCC): ICI response proportions and Kaplan–Meier OS curves by PIANOS risk group. **F** PRJEB25780 (RCC): ICI response proportions by PIANOS risk group. **G** NIHMS1737783 (RCC): ICI response proportions and violin plot showing

difference of CD8+ cells ($p < 1 \times 10^{-15}$) and programmed death ligand-1 (PD-L1) expression ($p < 1 \times 10^{-15}$) per tumor area by immunohistochemistry (IHC) staining image and CD247 expression ($p < 1 \times 10^{-15}$) by PIANOS risk group. *P* values for Kaplan–Meier curves from two-sided log-rank test. *P* values for bar plot proportions from a two-sided Chi-square test. *P* values for violin plots from the two-sided Wilcoxon rank-sum test. For violin/box plots (**G**): median (center), 25th/75th percentiles (box), whiskers (1.5xIQR, outliers not shown). SKCM skin cutaneous melanoma, NSCLC non-small cell lung cancer, UC urothelial carcinoma, RCC renal cell carcinoma. Source data are provided as a Source Data file.

differing prognoses, only patients with a poor prognosis (recurrence within 3 years, $n = 190$) or a relatively good prognosis (no recurrence beyond 5 years, $n = 174$) were selected to train the classification model. First, we transformed gene expression data into pathway enrichment scores based on ssGSEA. All gene sets from the MSigDB database version 7.0[94] were used for analysis. Subsequently, we randomly selected 70% of the samples to train the k-TSP model, and the remaining 30% were used for testing. After repeating 100 times, 100 models for prognosis classification were obtained, and k-TSPs with balanced accuracy higher than 0.6 in the test set were retained and constructed as PIANOS. If more than half of the k-TSPs predicted the patient to be at high risk, the patient was assigned to the high-risk group; otherwise, it was assigned to the low-risk group. All parameters were fixed after training, and risk groups for new patients were predicted using the confirmed model. In developing our personalized risk assessment model for colorectal cancer, the primary objective was to create a broadly applicable prognostic tool. Consequently, the model construction did not incorporate sex-specific analyses.

### RNA-seq and WGS collection and sequencing

WGS: Genomic DNA (1 μg starting material) was physically fragmented using a Covaris E220 instrument. Fragments of 200–400 bp were selected using AMPure XP beads. Library construction for WGS was performed using established in-house protocols with individual reagents rather than a commercial kit, involving end-repair, 3′ adenylation, adapter ligation, and PCR amplification.

RNA-seq: Ribosomal RNA (rRNA) was removed from total RNA using the MGIEasy rRNA Depletion Kit (32 RXN, Cat. No.: 1000005953, MGI). Subsequently, the rRNA-depleted RNA fraction was fragmented and used for first-strand cDNA synthesis via random hexamer-primed reverse transcription, followed by second-strand cDNA synthesis using a conventional random primer method. RNA Index Adapters were added after end-repair and A-tailing.

Sequencing and Adapter Trimming: Both WGS and RNA-seq libraries were subjected to paired-end sequencing (100 bp + 100 bp + 10 bp reads) on a DNBSEQ-T1 sequencer (MGI Tech, Shenzhen, Guangdong, China). Adapter sequences used for filtering were identical for both RNA and WGS data: Filter_Adp5: AAGTCG GATCGTAGCCATGTCGTTCTGTGAGCCAAGGAGTTG Filter_Adp3: AAGTCGGAGGCCAAGCGGTCTTAGGAAGACAA

### Evaluating the performance of PIANOS

The performance of PIANOS was evaluated using C-index, D-index, and Kaplan–Meier curves. C-index is the proportion of all paired patients whose predicted results are consistent with actual results and is the most frequently used evaluation indicator for survival models. D-index is an estimate of the traditional hazard ratio (HR). We used the survcomp R package to calculate C-index and D-index. Kaplan–Meier method estimates the probability of survival of patients in different risk groups at a specific time and calculates the corresponding $P$ value using the log-rank test. We performed univariate and multivariate analyses using the survival package for CIT, TCGA, and COCC, and univariate analyses for other GEO cohorts. We used the pROC package to assess the predictive performance of PIANOS, SIA, iCMS, CMS, and CRPS for colorectal cancer recurrence. Specifically, SIA, iCMS, and CRPS scores were computed for the study cohorts following the calculation methods described in their original publications. SIA was estimated based on the ratio of CD8A to C1QA gene expression. iCMS was calculated using the nearest template prediction method, which leverages the expression pattern template of iCMS marker genes (Supplementary Data 13). The CRPS values were obtained using the code available at https://github.com/SkymayBlue/U-CAN_CRPS_Model. Classification systems were dichotomized into two groups based on established risk stratifications described in the original publications. Specifically, CRPS was divided into CRPS4 versus others,

and CMS was divided into CMS4 versus others. For SIA, which yields a continuous score, we directly used the calculated SIA value for ROC curve analysis without dichotomization. Given that most recurrences from CRC occur within the first 3 years post-surgery, we benchmarked PIANOS and these established methods based on their ability to predict 3-year recurrence. It is important to acknowledge that several of these benchmark methods were originally developed and validated using different outcome metrics and follow-up durations. Specifically, the original CMS publication validated its subtypes against 72-month overall survival, relapse-free survival (RFS), and survival after relapse. The iCMS subtyping was validated against 150-month RFS, OS, and survival after relapse. The SIA signature was validated for RFS and OS over the longest possible follow-up period in its original study, and the CRPS model was validated for 60-month RFS and OS. To test the stability of PIANOS predictions, we randomly reduced the number of input genes to evaluate changes in C-index and D-index. We compiled and evaluated the predictive performance of 105 published models of CRC using the C-index across several of our main cohorts. Each model was scored based on relevant features, such as gene expression patterns (an overview of parameter settings is provided in Supplementary Data 14), and these scores were used as inputs to calculate the C-index. Specifically, the model by Tokunaga employed a random forest approach, while Tokunaga used the ssGSEA method. All other models used a weighted summation approach for scoring. The code for these analyses has been uploaded to GitHub, and further details can be found in the Data and code availability section. None of the cohorts were standardized.

### Landscape of difference in PIANOS groups

We used the maftools package[95] to analyze gene mutation data from TCGA, COCC, and ACICAM. The results were visually presented using waterfall plots that displayed the frequency of mutations in the top 15 genes within each PIANOS subgroup across these cohorts. Subsequently, we aggregated mutation frequencies across three cohorts and ascertained whether the mutation frequencies of the high- and low-risk group were significantly different using chi-square tests.

The calculation method for the HALLMARK pathway enrichment score was the same as before. Differential pathway analysis and differential gene expression analysis were conducted using the LIMMA package[96], in which selected genes with |logFC|>1 and $P$ value < 0.05 were deemed differentially expressed genes.

### Treatment response-related analysis

Gene set enrichment analysis (GSEA) was performed using the GSEA function from the clusterProfiler package[97], with gene ranking based on the fold change values obtained from differential expression analysis using the limma package. The GSEA function was then applied, and $p$ value adjustment was conducted using the Benjamini–Hochberg (BH) method to correct for multiple hypothesis testing. Specific code used in these analyses is available in the Data and Code Availability section. We used the genomics of drug sensitivity in cancer (GDSC) database to predict sensitivity to common chemotherapeutic agents for colorectal cancer. To analyze immune-related characteristics within tumor samples, we used the xCell package[98] to estimate the abundance of various cells in different patient tumors and obtained data of endogenous and exogenous immune responses using the maftools package. The immune subtypes of TCGA patients were obtained from ref. 35. TIDE score is often used to evaluate sensitivity of immunotherapy, and we calculated TIDE score for each patient using online tools (http://tide.dfci.harvard.edu/).

### Morphological and Imaging of COCC

CT scans of all patients were obtained from COCC at the Sixth Affiliated Hospital of Sun Yat-sen University. We selected preoperative and postoperative CT images of patients with complete imaging follow-up

information, retrieved DICOM files, and extracted images at the tumor level for display.

Formalin-fixed paraffin-embedded (FFPE) tumor tissue blocks were cut into 4-μm sections for H&E and IHC staining. Paraffin-embedded tumor tissue sections were dried at 65 °C for 15 min and then placed in an automatic IHC machine (BenchMark XT, Roche) for staining. The following primary antibodies were used: MLH1 (mouse monoclonal, clone ES05, Cat No. MAB-0789, MXB), MSH2 (rabbit monoclonal, clone LBP2-MSH2, Cat No. IR376, LBP), MSH6 (rabbit monoclonal, clone EP49, Cat No. ZA-0541, ZSGB-BIO), PMS2 (rabbit monoclonal, clone EP51, Cat No. ZA-0542, ZSGB-BIO), and Ki67 (mouse monoclonal, clone MIB-1, Cat No. IR62661-2, Dako). All antibodies were ready-to-use formulations and applied according to the manufacturer's instructions within the automated system. The positive signals presented as a brownish-yellow color localized in the nuclei. All slides were reviewed by a pathologist who confirmed the presence of tumor areas. Whole slide images (WSIs) were acquired at a magnification of 40× on an Aperio scanner.

### Single-cell analysis

We used the Seurat package to perform single-cell analysis. Expression data of GSE178341 was downloaded from Broad Institute's Single Cell Portal (https://singlecell.broadinstitute.org/single_cell/study/SCP1162). All parameters were set the same as the original analysis. To predict each patient's PIANOS grouping, we aggregated the gene expression data across all cell types for that individual, effectively creating a patient-level gene expression profile. The resulting patient-specific PIANOS groupings were then employed for subsequent downstream analyses.

### Statistics and reproducibility

The general study design involved retrospective cohort analysis. Sample sizes for the cohorts used in this study were primarily determined by the availability of public datasets or the number of patients enrolled in our in-house COCC cohort during the study period. No statistical method was used to predetermine sample size. Patients with mismatched gene expression data and clinical information were excluded from the analyses to ensure data integrity; otherwise, no data were excluded from the analyses beyond standard quality control measures outlined for specific datasets. The experiments were not randomized.

Figures were generated using R package ggplot2. The world map data were sourced from maps package. The schematic diagrams in Fig. 8B, D, E, G were drawn using Figdraw.

Differences in continuous values between two groups were examined using a two-sided Wilcoxon rank-sum test. Differences in categorical variables were examined using the chi-square test. To test for a significant association between multiple categorical variables, we used the chi-square test. Pearson's correlation analysis was used to compare clinical characteristics. The calculation of C-index and D-index, as well as model comparison were conducted by the survcomp package[99]. A hypergeometric test was used to evaluate the association between categorical factors. Statistical significance was set at $P < 0.05$. All analyses were performed using the R language (version 4.3.1).

### Reporting summary

Further information on research design is available in the Nature Portfolio Reporting Summary linked to this article.

### Data availability

Gene data and clinical information from all public datasets are available online, with the URLs mentioned in the Supplementary Data 31, including ACICAM [https://www.ncbi.nlm.nih.gov/pmc/articles/PMC10202816/]: Datasets of RNA-Seq, whole-exome sequencing, TCR sequencing of 348 CRC samples and adjacent normal tissue samples, TCGA [https://portal.gdc.cancer.gov/]: Datasets of RNA-seq, whole-genome sequencing of 633 CRC samples and adjacent normal tissue samples, IMvigor210 [https://pubmed.ncbi.nlm.nih.gov/29443960/]: RNA-seq of 298 pretreatment tumor tissue samples from metastatic urothelial carcinoma patients treated with atezolizumab, NIHMS1737783 [https://www.ncbi.nlm.nih.gov/pmc/articles/PMC8493486/]: Whole-exome sequencing and mRNA-seq of 886 baseline tumor tissue samples from advanced renal cell carcinoma patients, PRJEB23709 [https://linkinghub.elsevier.com/retrieve/pii/S1535-6108(19)30037-6]: RNA-seq of 158 tumor biopsy specimens from metastatic melanoma patients, PRJEB25780 [https://www.nature.com/articles/s41591-018-0101-z]: RNA-seq of 61 pretreatment tumor tissue samples from metastatic gastric cancer patients treated with pembrolizumab, GSE104645: Expression profiling by array of 193 formalin-fixed, paraffin-embedded primary colorectal cancer tumor samples, GSE31595: Expression profiling by array of 37 stage II and III colon cancer tumor samples, GSE39582: Expression profiling by array of 585 colorectal tissue samples, GSE35640: Expression profiling by array of 65 pretreatment tumor biopsy samples from metastatic melanoma and early-stage non-small-cell lung cancer patients, GSE14333: Expression profiling by array of 290 primary colorectal cancer tumor samples, GSE39084: Expression profiling by array of 70 primary colorectal cancer tumor tissue samples, GSE28722: Expression profiling by two-color microarray of 129 primary colorectal tumor tissue samples hybridized against a pooled common reference, GSE41258: Expression profiling by array of 390 colorectal patient tissue samples, GSE178341: Single-cell RNA-seq of 371,223 dissociated cells from primary, treatment-naïve colorectal cancer tumors and adjacent normal mucosa, spanning 28 mismatch-repair proficient and 34 mismatch-repair deficient patients, GSE17538: Expression profiling by array of 232 primary colorectal cancer tumor samples, GSE38832: Expression profiling by array of 122 primary colorectal cancer tumor tissue samples, GSE63624: Expression profiling by exon microarray of 52 primary proximal colon cancer tumor tissue samples, GSE75315: Expression profiling by exon microarray of 211 primary colorectal cancer tumor tissue samples, GSE136961: Targeted RNA sequencing of 21 pretreatment non-small cell lung cancer tumor tissue samples, GSE91061: RNA-seq of 118 tumor biopsy samples from 65 melanoma patients, GSE143985: Expression profiling by array of 91 primary colorectal tumor samples and GSE87211: Expression profiling by array of 363 rectal cancer and matched mucosa samples from 243 patients. The sequencing data and associated clinical information for the COCC cohort generated in this study have been deposited in the China National Center for Bioinformation (CNCB) BioProject database under accession code HRA007315: Datasets of genome, transcriptome, epigenetics of 1050 CRC samples and adjacent normal tissue samples, DNBSEQ-T1×5RS. Due to patient privacy regulations and legal restrictions regarding the sharing of human genomic data, access to the COCC cohort data is available under restricted access. Access can be obtained by contacting the corresponding author, Dr. Xiaojian Wu(wuxjian@mail.sysu.edu.cn), with an appropriate data access agreement (Supplementary Note 1). Source data are provided as a Source Data file. Source data are provided with this paper.

### Code availability

All the codes were implemented in R (version 4.3.1). Packages used are available online, with the URLs mentioned in Supplementary Data 31. The scripts to replicate each step of results and plots can be accessed in a GitHub repository[100] (https://github.com/LidocaineQ/PIANOS and https://doi.org/10.5281/zenodo.15396780) under the Apache License 2.0.

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

## Acknowledgements

This study was supported by Noncommunicable Chronic Diseases-National Science and Technology Major Project (2023ZD0501600, F.G.), Guangzhou Key Research and Development Project (No.202206080008, X.W.), Guangdong Key Research and Development Project (No.2023B1111040003, X.W.), Guangdong Basic and Applied Basic Research Foundation (No.2023B1515130008, X.W.), Tian-shan Talents · Leading Medical Talents in Guangdong Province Cooperative Expert Studio (NO.KSYJ2022001, X.W.), National Natural Science Foundation of China (No.8227242, F.G.), Guangzhou Basic and Applied Basic Research Fund (No.2024A04J9983, F.G.), Guangdong Basic and Applied Basic Research Foundation (No.2023A1515110196, D.C.), Postdoctoral Fellowship Program of CPSF (No.GZC20233214, D.C.), the program of Guangdong Provincial Clinical Research Center for Digestive Diseases (2020B1111170004, X.W.).

## Author contributions

D.C., H.Q., Y.M., F.G., and X.W. conceived the ideas and designed the analysis. C.L., C.H., B.G., and H.Q. collected the data. D.C., H.Q., Q.Y., H.L., F.G., and Y.M. performed bioinformatics data analysis. D.C., H.Q., Q.Y., H.L., X.Z., and Y.M. performed clinical data analysis. D.C., H.Q., Y.M., F.G., and X.W. wrote the paper.

## Competing interests

The authors declare no competing interests.
