## [Peer Review File · Nature Communications]

Personalized Risk Stratification in Colorectal Cancer via PIANOS System

Corresponding Author: Professor Xiaojian Wu

Version 0:

Reviewer comments:

Reviewer #1

(Remarks to the Author)
Comments have been address.

(Remarks on code availability)

Reviewer #3

(Remarks to the Author)
Cai et al seek to develop a transcriptomic risk prediction system for CRC. The developed predictor performs well when benchmarked to other transcriptomic predictors (Fig 3K).

What are the most important transcripts driving PIANOS and are they found also in other predictors?

Can performance be benchmarked to mIHC scores such as ImmunoScore and SIA?

How were the particular cohorts used in the study selected? Do they represent clinical trial populations or real world populations?

Figure 3D should include also the iCMS system and the CRPS system (Nunes et al, 2024) for comparison as the latter is prognostic.

Figure 4A appears pointless to me and is not really discussed in the text. The genes shown include several known non-driver genes with high background mutation such as SYNE1, TTN, OBSCN, RYR2. What are the authors trying to convey with the figure?

From a clinical point of view, the associations between OS and bevacizumab use are questionable. The 30 patients either had met surgeries (details unknown) or have attained CR (on what treatment?) and then relapsed and received some combination of chemo and bevacizumab. To make conclusions about response prediction to bevacizumab in this small cohort is of no value.

(Remarks on code availability)

Version 1:

Reviewer comments:

Reviewer #3

(Remarks to the Author)

My comments and those of referee #2 have been sufficiently addressed.

The outcome used in benchmarking of PIANOS to other methods is 3-year recurrence, and several of the benchmark methods were developed based on 5-year OS/RFS. I think it would make sense to tabulate information on timepoint and outcome metric that each respective benchmark method is supposed to be informative for in the manuscript as there is otherwise a risk of comparison of apples and pears.

(Remarks on code availability)

RESPONSES TO REVIEWERS

We thank all the reviewers for their thorough evaluation of our manuscript. We are particularly grateful for the insightful comments provided by Reviewers 1 and 2 in the previous round of review. We believe these changes, guided by their expertise, have substantially enhanced the quality of the manuscript.

We are grateful to Reviewer 3 for the careful reading of the manuscript and insightful comments, which has prompted us to consider several important aspects of our work in more detail. Our responses to each of these comments are provided below. To facilitate the review process, modifications have been clearly marked in the revised manuscript. Deleted text is indicated with red strikethrough, and newly added text is highlighted in yellow.

Reviewer 3

Comment 1

What are the most important transcripts driving PIANOS and are they found also in other predictors?

Response:

Thank you for raising the important question of which transcripts are most influential in the PIANOS model and how they relate to those found in other published predictors. This is a crucial point for understanding the biological underpinnings of our model and its relationship to existing knowledge.

Due to the resampling procedure used in its construction, PIANOS comprises 35 distinct k-TSP submodels. Each submodel may capture different, yet complementary, pathway-level information relevant to CRC prognosis. In our original submission, we prioritized identifying frequently recurrent pathways across these 35 submodels (as shown in the Supplementary Figure 5). Our rationale was that pathways consistently appearing across multiple submodels likely represent robust and biologically significant contributors to CRC prognosis.

In direct response to your question, we have now performed a new analysis to identify the individual genes that appear most frequently across all pathways in PIANOS. This provides a gene-centric view of the key drivers within PIANOS, complementing the pathway-level analysis. We have updated Supplementary Figure 5 to include this new information, clearly showing both the

frequently recurring pathways and the frequently occurring genes. We have compiled a comprehensive comparison (provided in Comment Table 1) of the high-frequency genes identified in PIANOS with those reported in previously published colorectal cancer prognostic signatures. This comparison demonstrates the consistency of PIANOS with existing knowledge and may identify potentially novel prognostic markers for further study.

We believe this expanded analysis, incorporating both pathway-level recurrence and gene-level frequency, coupled with a direct comparison to other predictors, provides a much clearer and more complete answer to the reviewer's question. It demonstrates that PIANOS captures known prognostic information while also potentially identifying novel contributors to CRC outcome. We thank you again for this valuable suggestion, which has significantly strengthened our manuscript.

Supplementary Figure 5A, B Highest frequency pathways and genes identified in PIANOS.

We also revised this in the manuscript (Results section):

159 immune-related processes (Supplementary Figure 5A). To identify the most critical
 160 transcriptomic signals within PIANOS, we ranked the genes from these pathways based on their
 161 frequency of occurrence (Supplementary Figure 5B). Notably, genes such as DKK1, GZMB,
 162 THBS1, and CCL5—recognized for their key contributions to colorectal cancer prognosis—
 163 have been incorporated into several existing prognostic models²⁴⁻³¹. We then examined

Comment table 1 Genes reported in published CRC prognostic models.

Gene Symbol	Full Name	Model name	PMID
GZMB	Granzyme B	Dai-MO	29377588
DKK1	Dickkopf-1	Goeman	33377637
DKK1	Dickkopf-1	Hao	20077526
DKK1	Dickkopf-1	Liang	34169901
GZMB	Granzyme B	Mao	33732694
DKK1	Dickkopf-1	Mo-FICDB	34211976
THBS1	Thrombospondin-1	Zhang-Med	32569190
CCL5	C-C chemokine ligand 5	Tokunaga	32191346

Comment 2

Can performance be benchmarked to mIHC scores such as ImmunoScore and SIA?

Response:

Thank you for your valuable suggestion to benchmark PIANOS's performance against established mIHC-based scores, specifically ImmunoScore and Stromal Immune Angle (SIA). Benchmarking against such clinically relevant metrics is crucial for evaluating the potential utility of our model.

We fully agree that comparing PIANOS to ImmunoScore, a well-validated and commercially available test, would be highly informative. We extensively investigated the possibility of obtaining ImmunoScore data for our cohort by exploring potential collaborations with institutions that routinely use ImmunoScore, and searching publicly available datasets for cohorts with ImmunoScore data for a meaningful comparison with our PIANOS. Unfortunately, despite these efforts, we were unable to access ImmunoScore data for our specific cohort. There is currently no authorized distributor of ImmunoScore in mainland China, and hurdles involved in obtaining the necessary data or performing the assay retrospectively on our samples proved insurmountable within the timeframe of this revision. We acknowledge this as a limitation and will actively pursue this comparison in future studies, should access to ImmunoScore become available.

On the other hand, we calculated the SIA for our cohort based on the published methodology. We then performed a direct comparison of the prognostic performance of SIA, PIANOS, CRPS, CMS and iCMS in predicting 3-year recurrence in our CRC cohort. We also performed survival

analysis on different subgroups defined by iCMS, CRPS, and SIA within the COCC cohort (presented in the updated Supplementary Figure 3). We are pleased to report that our results are consistent with the original publications, confirming the effectiveness of these systems and demonstrating the representativeness of our cohort (presented in the updated Supplementary Figure 4).

We believe that the comparison with SIA, CRPS, CMS and iCMS, along with our detailed explanation of the challenges in obtaining ImmunoScore data, provides response to your suggestion. We are committed to ongoing validation and benchmarking of PIANOS and will continue to explore opportunities for comparison with established clinical tools. We greatly appreciate the reviewer's insightful suggestion, which has helped us to contextualize the performance of our model.

Supplementary Figure 3F-J AUC for predicting 3-year recurrence comparing the performance of PIANOS, iCMS, SIA, CMS and CRPS among patients in CIT, TCGA, COCC, ACICAM and Meta-GEO cohorts.

Supplementary Figure 4 Kaplan-Meier curves for DFS based on the CMS, SIA, iCMS and CRPS in COCC. P values were calculated with log-rank test.

We also revised this in the manuscript (Introduction and Results section):

78 pathology⁹⁻¹³. For instance, the pathology-based SIA¹⁴, the iCMS developed through the
 79 integration of single-cell and transcriptomic analyses¹⁵, and the CRPS—which converts
 80 transcriptomic data into pathway enrichment scores and uses deep learning for modeling—
 81 have all been explored. However, these biomarkers frequently face accuracy and stability

135 To compare the predictive performance of PIANOS with that of existing colorectal cancer
136 prognostic systems including SIA, iCMS, CMS and CRPS, we evaluated the ROC curves for
137 3-year recurrence across different cohorts (Supplementary Figure 3F-J), revealing that PIANOS
138 exhibited the highest predictive capability in all cohorts. To further contextualize our findings,
139 we performed survival analysis on the different subtypes defined by the CMS, SIA, iCMS, and
140 CRPS classification systems within the COCC cohort. Importantly, the prognostic stratification
141 observed within our COCC cohort for these classification systems was consistent with their
142 previously reported prognostic performance (Supplementary Figure 4A-D). To investigate

676 14 → Mezheyeuski, A. *et al.* An immune score reflecting pro- and anti-tumoural balance of tumour
677 microenvironment has major prognostic impact and predicts immunotherapy response in solid cancers.
678 *EBioMedicine* **88**, 104452, doi:10.1016/j.ebiom.2023.104452 (2023).⁴¹
679 15 → Joanito, I. *et al.* Single-cell and bulk transcriptome sequencing identifies two epithelial tumor cell states
680 and refines the consensus molecular classification of colorectal cancer. *Nat Genet* **54**, 963-975,
681 doi:10.1038/s41588-022-01100-4 (2022).⁴²
682 16 → Nunes, L. *et al.* Prognostic genome and transcriptome signatures in colorectal cancers. *Nature* **633**, 137-
683 146, doi:10.1038/s41586-024-07769-3 (2024).⁴³

Comment 3

How were the particular cohorts used in the study selected? Do they represent clinical trial populations or real-world populations?

Response:

We appreciate the reviewer's important question regarding the selection criteria and representativeness of the cohorts used in our study. It is crucial to establish whether these cohorts reflect real-world clinical populations or are limited to specific clinical trial settings. We have clarified the cohort selection process in the Methods section of the revised manuscript.

All cohorts included in this study are retrospective and represent real-world patient populations, not specifically selected clinical trial populations. They were chosen based on the availability of both comprehensive transcriptomic data (RNA sequencing) and corresponding clinical follow-up information, including survival outcomes. The cohorts include publicly available datasets (TCGA, ACICAM and multiple datasets from the GEO) and our in-house COCC cohort. The COCC CRC cohort for independent validation was CRC subproject of ICGC-ARGO project, a key resource that our team has established and continues to develop. COCC comprises patients with pathologically confirmed colorectal cancer who underwent curative-intent resection at the Sixth Affiliated Hospital

of Sun Yat-sen University. All involved patients have complete transcriptomic data and clinical outcome data.

Our deliberate inclusion of cohorts from diverse geographical regions and different sequencing platforms was to enhance the generalizability of our findings. By demonstrating the robust performance of PIANOS across these varied datasets, we aim to show its potential applicability to a broad range of real-world colorectal cancer patients, rather than being limited to a specific clinical trial setting or a single institution's patient population. This heterogeneity increases the likelihood that our model can be successfully translated into clinical practice.

We believe this clarifies that our study utilizes real-world, retrospective cohorts representing diverse patient populations and sequencing platforms. This approach strengthens the evidence for the potential clinical utility and generalizability of the PIANOS model. We thank you again for raising this important point.

We also revised this in the manuscript (Methods section):

448 **Datasets**[↵]

449 **To construct and validate a cross-platform robust single-sample risk stratification system, our**

450 **research included cohorts from various countries and regions.** We obtained gene expression

451 profiles and follow-up data of 3666 patients with CRC^{34,68-83} to validate PIANOS, 396 patients

452 with CRC^{84,85} to predict neoadjuvant therapy efficacy, including data from the Gene Expression

453 Omnibus (GEO), The Cancer Genome Atlas (TCGA), and in-house Clinical Omics Study of

454 Colorectal Cancer in China (COCC) CRC cohorts. GEO cohorts were downloaded from GEO

455 website (<https://www.ncbi.nlm.nih.gov/geo/>) and pre-processed using GEOquery R package⁸⁶.

456 French multicenter cohort (CIT) served as training dataset while 10 GEO cohorts were merged

457 into Meta-GEO validation cohort. Transcripts Per Million (TPM) data and H&E slides from

458 COAD and READ TCGA cohorts were downloaded from (<https://portal.gdc.cancer.gov/>) and

459 merged into TCGA CRC validation cohort. The COCC CRC cohort for independent validation

460 was CRC subproject of ICGC-ARGO project (<https://www.icgc-argo.org/page/114/cgcc>), and

461 all cases were collected from The Sixth Affiliated Hospital of Sun Yat-sen University,

462 GuangZhou, China. **All patients underwent curative surgical resection, and the diagnosis of**

463 **colorectal cancer was rigorously confirmed by histopathological evaluation.** Transcriptome data

Comment 4

Figure 3D should include also the iCMS system and the CRPS system (Nunes et al, 2024) for comparison as the latter is prognostic.

Response:

Thank you for your insightful suggestion to include the iCMS and CRPS systems in the comparison presented in Figure 3D. We agree that comparing PIANOS to these established prognostic systems, particularly given CRPS's recent publication and demonstrated prognostic value, is essential for contextualizing our model's performance. We have expanded the analysis in Figure 3D to include both iCMS and CRPS alongside PIANOS and other relevant comparators (as mentioned in Comment 2). This revised figure now directly compares the time-dependent AUCs for all these systems across the datasets analyzed (as updated in Supplementary Figure 3, 4).

We believe that this direct comparison of PIANOS with iCMS and CRPS, within the context of time-dependent AUC analysis for recurrence prediction, significantly strengthens the manuscript. It provides a more comprehensive evaluation of PIANOS's performance relative to other state-of-the-art prognostic systems. We are grateful for your suggestion, which has helped us to improve the rigor and clarity of our findings.

Supplementary Figure 3F-J AUC for predicting 3-year recurrence comparing performance of PIANOS, iCMS, SIA, CMS and CRPS in CIT, TCGA, COCC, ACICAM and Meta-GEO cohorts.

Supplementary Figure 4 Kaplan-Meier curves for DFS based on the CMS, SIA, iCMS and CRPS in COCC. P values were calculated with log-rank test.

We also revised this in the manuscript (Results section):

135 To compare the predictive performance of PIANOS with that of existing colorectal cancer
 136 prognostic systems including SIA, iCMS, CMS and CRPS, we evaluated the ROC curves for
 137 3-year recurrence across different cohorts (Supplementary Figure 3F-J), revealing that PIANOS
 138 exhibited the highest predictive capability in all cohorts. To further contextualize our findings,
 139 we performed survival analysis on the different subtypes defined by the CMS, SIA, iCMS, and
 140 CRPS classification systems within the COCC cohort. Importantly, the prognostic stratification
 141 observed within our COCC cohort for these classification systems was consistent with their
 142 previously reported prognostic performance (Supplementary Figure 4A-D). To investigate

Comment 5

Figure 4A appears pointless to me and is not really discussed in the text. The genes shown include several known non-driver genes with high background mutation such as SYNE1, TTN, OBSCN, RYR2. What are the authors trying to convey with the figure?

Response:

We sincerely appreciate your insightful comment regarding Figure 4A and its lack of discussion in the original manuscript. We recognize that the original presentation of the figure, showcasing the most frequently mutated genes in the COCC cohort, did not sufficiently convey its intended purpose and included genes (like SYNE1, TTN, OBSCN, and RYR2) known to have high background mutation rates and potentially not be driver genes. You rightly questioned the message we were trying to convey.

We agree that simply presenting the most frequently mutated genes without context was insufficient. We have performed a new, comparative analysis focusing on the differential mutation frequencies between the high-risk and low-risk groups defined by our PIANOS model. This new analysis, presented as Supplementary Figure 6D, directly compares the mutation profiles and highlights statistically significant differences. This approach allows us to identify mutations that are enriched in one risk group versus the other, potentially pointing to genes that contribute to the distinct phenotypes.

We have significantly expanded the discussion of these mutational differences in the main text (Results section). We now explicitly address the potential functional implications of the differentially mutated genes, drawing upon existing literature. Specifically, we discuss: How higher-frequency mutations in SMAD4 and TP53 in the high-risk group might be linked to enhanced EMT and angiogenesis, consistent with the observed phenotype. How wild-type TP53 in the low-risk group could contribute to a more robust immune phenotype. The potential roles of mutations in genes like TCF7L2, FAT4, MUC16, and RYR2 (even those with high background mutation rates) in modulating immune activity and influencing prognosis in CRC and other cancers, citing relevant studies where these genes have shown functional significance in specific contexts.

We believe that these revisions directly address your concerns and substantially strengthen the manuscript. The mutational analysis now provides valuable insights into the potential molecular

mechanisms underlying the different risk profiles identified by our model. Thank you again for your valuable feedback, which has helped us improve the rigor and clarity of our work.

Supplementary Figure 6D High-frequency mutated genes with differential frequencies between COCC high- and low-risk groups.

We also revised this in the manuscript (Discussion section):

333 responsiveness to immunotherapy in solid tumors harboring MUC16 mutations^{43,44}. In the
 334 COCC cohort, mutation analysis demonstrated that the high-risk group exhibited an elevated
 335 mutation frequency of SMAD4 and TP53. Previous studies have indicated that these features
 336 are associated with enhanced EMT and angiogenesis^{45,46}, whereas wild-type TP53 may confer
 337 a more robust immune phenotype⁴⁷. Moreover, mutations in TCF7L2, FAT4, MUC16, and
 338 RYR2 have been correlated with heightened immune activity and improved prognosis in CRC
 339 and other malignancies⁴⁸⁻⁵¹. Through integration of differential gene expression and survival

Comment 6

From a clinical point of view, the associations between OS and bevacizumab use are questionable. The 30 patients either had met surgeries (details unknown) or have attained CR (on what treatment?) and then relapsed and received some combination of chemo and bevacizumab. To make conclusions about response prediction to bevacizumab in this small cohort is of no value.

Response:

We sincerely appreciate your insightful feedback and the opportunity to refine our manuscript. We acknowledge your concerns regarding the interpretation of bevacizumab response in our study,

particularly given the small size and retrospective nature of the sub-cohort. We have made significant revisions to address these points, as detailed below:

In direct response to your comment, we have revised the subtitle to: “Colorectal Cancer Patients in the PIANOS High-Risk Group Exhibit Higher Angiogenesis Activation Features.” This change shifts the emphasis from a definitive statement on treatment sensitivity to a more accurate reflection of our findings, which center on the observed angiogenic characteristics of the high-risk group.

We have carefully toned down our statements regarding bevacizumab sensitivity. The revised text now explicitly states that, while our small retrospective sub-cohort suggests a potential trend toward improved outcomes with bevacizumab plus chemotherapy in high-risk patients (characterized by activated angiogenic features), these findings are preliminary and require further validation. We have removed any language that could be interpreted as definitive conclusions about treatment prediction.

We agree that further clarification of the bevacizumab-treated cohort was needed. The revised manuscript now clearly states that the bevacizumab-treated sub-cohort comprised patients with metastatic colorectal cancer who underwent standard curative resection. Following tumor recurrence, these patients received combination therapy involving bevacizumab and chemotherapy. Overall survival (OS) was the endpoint used in our survival analysis.

As you rightly pointed out, drawing conclusions about treatment implications from a cohort of 30 patients is not sufficiently rigorous. We wholeheartedly agree. To mitigate potential selection bias and enhance the robustness of our results, we implemented 1:1 Propensity Score Matching (PSM) within the stage IV patient group, based on key variables including age, sex, and tumor location. While PSM helped to balance the groups based on the selected covariates, the bevacizumab findings still stem from a relatively small, retrospective cohort, which inherently introduces limitations. To address this directly, we have expanded the limitations section to specifically highlight the constraints of the bevacizumab-related analysis. We now explicitly state: The bevacizumab findings are derived from a limited, retrospective cohort. While our model effectively personalizes risk classification and offers some insight into potential treatment inclinations, the observed trends related to bevacizumab require confirmation. Prospective studies with larger, well-defined cohorts are essential to validate both the model's overall predictive performance and the

specific trends observed in drug sensitivity.

We deeply appreciate your valuable feedback, which has significantly strengthened the rigor and clarity of our manuscript. We believe these revisions ensure that our conclusions are appropriately cautious and that the limitations of the bevacizumab analysis are transparently acknowledged. We hope that these changes, driven by your insightful suggestions, make our article more rigorous and suitable for publication in Nature Communications.

We also revised this in the manuscript

Results section:

214 ~~Metastatic CRC patients with high PIANOS scores were more sensitive to bevacizumab~~[←]
215 **Metastatic CRC patients with high PIANOS scores exhibit higher angiogenesis activation**
216 **features**[←]
217 ~~Enrichment analysis revealed enhanced angiogenesis pathway in high-risk group (Figure 4E),~~
218 ~~suggesting potentially increased sensitivity to bevacizumab, a widely utilized monoclonal~~
219 ~~antibody in CRC therapy.~~ **Enrichment analysis revealed an enhanced angiogenesis pathway in**
220 **the high-risk group (Figure 4E), indicating that bevacizumab, a widely utilized monoclonal**
221 **antibody in CRC therapy, might have potential clinical utility in this subgroup**³³. We confirmed

Methods section:

471 recurrence or death. ~~The bevacizumab-treated sub-cohort (n=30) from COCC included only~~
472 ~~metastatic CRC patients who experienced recurrence after standard surgical treatment and~~
473 ~~subsequently received combined chemotherapy and bevacizumab treatment. Therefore, overall~~
474 ~~survival (OS), defined as duration from start of treatment until death from any cause, was used~~
475 ~~as endpoint. In survival analysis of high- and low-risk groups, we used Propensity Score~~
476 ~~Matching (PSM) with 1:1 matching for age, sex, and location to reduce selection bias and~~
477 ~~improve result accuracy.~~ The bevacizumab-treated subgroup comprised metastatic colorectal
478 cancer patients who underwent standard curative resection and, following tumor recurrence,
479 received combination therapy with bevacizumab and chemotherapy. Accordingly, overall
480 survival (OS), defined as duration from start of treatment until death from any cause or the last
481 follow-up, was employed as the endpoint in our survival analysis. To reduce potential selection
482 bias and improve result accuracy, we employed 1:1 Propensity Score Matching (PSM) in stage
483 IV patients based on variables including age, sex, and tumor location.[↵]

Discussion section:

370 remains inconsistent across different populations and assays. ~~Our research identified~~
371 ~~characteristics of low-risk patients, including elevated VEGFA expression, active angiogenic~~
372 ~~pathways, and increased endothelial cell numbers expressing high levels of VEGFA, indicating~~
373 ~~intense angiogenic activity. Analysis of our in-house cohort revealed substantial benefits from~~
374 ~~bevacizumab treatment only in high-risk group. Based on this evidence, we propose that~~
375 ~~PIANOS can serve as a reliable predictive marker in future clinical trials focused on~~
376 ~~angiogenesis-targeting therapies, helping personalize treatment more precisely and effectively~~
377 ~~for patients.~~ Our study identified that high-risk patients exhibited elevated VEGFA expression,
378 enhanced angiogenic pathway activity, and a greater abundance of endothelial cells expressing
379 high VEGFA levels—features that collectively indicate robust angiogenic activation.
380 Encouragingly, our analysis of a small cohort of stage IV patients demonstrated a promising
381 trend toward improved overall survival in the high-risk group receiving bevacizumab compared
382 to those not treated. Although these preliminary findings derived using PSM support the
383 potential predictive value of the PIANOS model, larger and more rigorously designed clinical
384 studies are essential to confirm these observations and to further establish its clinical utility.[↵]

412 ~~implemented. Although our model effectively personalizes risk classification and provides~~
413 ~~practical treatment decision-making guidance for various patient groups, it remains a~~
414 ~~retrospective study. Future studies should validate the effectiveness of PAINOS when samples~~
415 ~~are collected prospectively. Drug sensitivity prediction cohort is retrospective, necessitating~~
416 ~~further prospective clinical trials to confirm its predictive performance.~~ Although our model
417 effectively personalizes risk classification and provides some potential treatment inclinations,
418 our study is retrospective, and the bevacizumab-related analysis is based on a limited small-
419 scale cohort. Therefore, prospective studies in larger cohorts are required to verify the predictive
420 performance of the model and the drug sensitivity of the stratified groups. Additionally, Further

425 CRC. Patients in low-risk group may benefit from chemotherapy and immunotherapy regimens,
426 whereas those in high-risk group ~~may benefit from bevacizumab monotherapy and aggressive~~
427 ~~chemotherapy~~ exhibit activated angiogenic features and may be more sensitive to bevacizumab
428 combined with chemotherapy. These results indicate that PIANOS not only serves as a
429 complementary tool for TNM staging system but also enhances application of precise treatment
430 strategies. ↵

RESPONSES TO REVIEWERS

Reviewer #3

My comments and those of referee #2 have been sufficiently addressed.

The outcome used in benchmarking of PIANOS to other methods is 3-year recurrence, and several of the benchmark methods were developed based on 5-year OS/RFS. I think it would make sense to tabulate information on timepoint and outcome metric that each respective benchmark method is supposed to be informative for in the manuscript as there is otherwise a risk of comparison of apples and pears.

Response:

Thank you for your insightful comments and for recognizing that our previous responses have sufficiently addressed your and Referee #2's points. We particularly appreciate your feedback regarding the benchmarking of PIANOS against other methods using a 3-year recurrence outcome, especially when several benchmark methods were originally developed based on different timepoints and outcome metrics (e.g., 5-year OS/RFS).

To address this specific concern and ensure clarity, we have now incorporated a detailed textual description into the manuscript. This new text is placed directly after the paragraph detailing the calculation of scores for the benchmark models. In this addition, we first provide a justification for using 3-year recurrence as our benchmarking endpoint. More importantly, we explicitly state the original outcome metric(s) and follow-up timepoint(s) for which each benchmark method (CMS, iCMS, SIA, and CRPS) was initially developed and validated. Additionally, we have updated Supplementary Figure 3 to include a direct performance comparison for 5-year DFS prediction between our PIANOS model and the other benchmark methods, and we have tabulated the detailed timepoints, outcome metrics, and corresponding AUC values in the Source Data file for this figure.

We believe this textual addition directly addresses your concern by providing the necessary context for the comparisons made, thus mitigating the risk of misinterpretation and ensuring readers are aware of the original design of the benchmarked methods.

We hope these revisions fully address your comment and meet with your approval.

calculated SIA value for ROC curve analysis without dichotomization. Given that most recurrences from CRC occur within the first three years post-surgery, we benchmarked PIANOS and these established methods based on their ability to predict 3-year recurrence. It is important to acknowledge that several of these benchmark methods were originally developed and validated using different outcome metrics and follow-up durations. Specifically, the original CMS publication validated its subtypes against 72-month overall survival, relapse-free survival (RFS), and survival after relapse. The iCMS subtyping was validated against 150-month RFS, OS, and survival after relapse. The SIA signature was validated for RFS and OS over the longest possible follow-up period in its original study, and the CRPS model was validated for 60-month RFS and OS. To test stability of PIANOS predictions, we randomly

Methods section

(Figure 3D-H, Supplementary Figure 3A-E). To compare the predictive performance of PIANOS with that of existing colorectal cancer prognostic systems including SIA, iCMS, CMS and CRPS, we evaluated the ROC curves for 3-year and 5-year recurrence across different cohorts (Supplementary Figure 3F-O), revealing that PIANOS exhibited the highest predictive capability in all cohorts. To further contextualize our findings, we performed survival analysis

Results section

Supplementary Figure 3

Supplementary Figure 3

	A	B	C	D	E
1	Supplementary Figure 3				
2	Models	Cohorts	AUC-DFS 3 years	AUC-DFS 5 years	
3	PIANOS	CIT	0.76	0.75	
4	PIANOS	TCGA	0.63	0.56	
5	PIANOS	COCC	0.69	0.71	
6	PIANOS	ACICAM	0.65	0.66	
7	PIANOS	Meta-GEO	0.68	0.68	
8	SIA	CIT	0.54	0.54	
9	SIA	TCGA	0.55	0.61	
10	SIA	COCC	0.56	0.57	
11	SIA	ACICAM	0.57	0.56	
12	SIA	Meta-GEO	0.54	0.53	
13	CRPS	CIT	0.57	0.54	
14	CRPS	TCGA	0.54	0.51	
15	CRPS	COCC	0.54	0.54	
16	CRPS	ACICAM	0.56	0.58	
17	CRPS	Meta-GEO	0.52	0.52	
18	iCMS	CIT	0.51	0.51	
19	iCMS	TCGA	0.54	0.59	
20	iCMS	COCC	0.55	0.54	
21	iCMS	ACICAM	0.51	0.52	
22	iCMS	Meta-GEO	0.51	0.5	
23	CMS	CIT	0.57	0.55	
24	CMS	TCGA	0.53	0.54	
25	CMS	COCC	0.56	0.55	
26	CMS	ACICAM	0.56	0.56	
27	CMS	Meta-GEO	0.54	0.57	

Source Data – Supplementary Figure 3